# Receptor tyrosine kinases CAD96CA and FGFR1 function as the cell membrane receptors of insect juvenile hormone

Yan-Xue Li, Xin-Le Kang, Yan-Li Li, Xiao-Pei Wang, Qiao Yan, Jin-Xing Wang, Xiao-Fan Zhao*

Shandong Provincial Key Laboratory of Animal Cells and Developmental Biology, School of Life Sciences, Shandong University, Shandong, China

## eLife Assessment

In this **important** study, Li and others identified cell membrane receptors for juvenile hormone (JH), a terpenoid hormone in insects that regulates their development and reproduction. While intracellular receptors for JH are well characterized, membrane receptors for JH have remained elusive for many years. The authors provide **convincing** evidence indicating that two receptor tyrosine kinases (RTKs), CAD96CA and FGFR1, modulate the genomic effects of JH by phosphorylating the intracellular receptors in the cotton bollworm, Helicoverpa armigera. Although differential functions of the two RTKs and potential effects of the other endogenous ligands of these RTKs on JH signaling remain unclear, this study lays a foundation for future studies.

*For correspondence:
xfzhao@sdu.edu.cn

Competing interest: The authors declare that no competing interests exist.

## Abstract

Juvenile hormone (JH) is important to maintain insect larval status; however, its cell membrane receptor has not been identified. Using the lepidopteran insect *Helicoverpa armigera* (cotton bollworm), a serious agricultural pest, as a model, we determined that receptor tyrosine kinases (RTKs) cadherin 96ca (CAD96CA) and fibroblast growth factor receptor homologue (FGFR1) function as JH cell membrane receptors by their roles in JH-regulated gene expression, larval status maintaining, rapid intracellular calcium increase, phosphorylation of JH intracellular receptor MET1 and cofactor Taiman, and high affinity to JH III. Gene knockout of *Cad96ca* and *Fgfr1* by CRISPR/Cas9 in embryo and knockdown in various insect cells, and overexpression of CAD96CA and FGFR1 in mammalian HEK-293T cells all supported CAD96CA and FGFR1 transmitting JH signal as JH cell membrane receptors.

## Introduction

Juvenile hormone (JH) plays a vital role in insect development and maintaining insect larval status. JH is an acyclic sesquiterpenoid known to enter cells freely via diffusion because of its lipid-soluble character (*Riddiford, 2020*). JH binds its intracellular receptor methoprene-tolerant protein (MET), a basic helix-loop-helix/Per-ARNT-SIM (bHLH-PAS) family protein (*Charles et al., 2011*; *Jindra et al., 2021*). MET forms a transcription complex with the transcription factor Taiman (TAI, also known as FISC, p160/SRC, and is a steroid receptor coactivator) to initiate gene transcription (*Charles et al., 2011*; *Zhu et al., 2003*). An important gene in the JH pathway is Krüppel homologue 1 (*Kr-h1*), which encodes the zinc-finger transcription factor *Kr-h1* (*Minakuchi et al., 2008b*; *Pecasse et al., 2000*; *Wu et al., 2021*). *Kr-h1* acts downstream of MET and is induced rapidly by JH to regulate larval growth and development (*Minakuchi et al., 2009*). Other genes, for example, the early trypsin gene of *Aedes aegypti* (*AaEt*; *Li et al., 2011*; *Noriega et al., 2003*), JH-inducible 21 kDa protein (*Jhp21*; *Zhang*

*et al., 1996*), JH esterase (*Jhe*; *Feng et al., 1999*; *Wroblewski et al., 1990*), vitellogenin (*Vg*; *Comas et al., 1999*; *Xu et al., 2014*), *Drosophila* JH-inducible gene 1 (*Jhi-1*), and JH-inducible gene 26 (*Jhi-26*; *Dubrovsky et al., 2000*) are regulated by JH.

However, some studies suggest that cell membrane receptors also play essential roles in JH signaling (*Davey, 2000*; *Jindra et al., 2021*). For example, in *A. aegypti*, receptor tyrosine kinases (RTKs) are involved in JH-induced rapid increases in inositol 1,4,5-trisphosphate, diacylglycerol, and intracellular calcium, leading to activation of calcium/calmodulin-dependent protein kinase II (CaMKII) to phosphorylate MET and TAI, resulting in *Kr-h1* gene transcription in response to JH (*Liu et al., 2015*). JH III, also via RTKs, leads to rapid calcium release and influx in *Helicoverpa armigera* epidermal cells (HaEpi cells) (*Wang et al., 2016*). JH induces MET phosphorylation to increase MET interacting with TAI, which enhances *Kr-h1* transcription in *H. armigera* (*Li et al., 2021*). In *Drosophila melanogaster*, JH through RTK and PKC protein kinase C (PKC) induces phosphorylation of ultraspiracle (USP) (*Gao et al., 2022*). The phenomenon that RTK transmits JH signal has long been predicted (*Liu et al., 2015*; *Ojani et al., 2016*); however, the RTKs critical for JH signaling have yet to be identified from numerous RTKs in vivo.

RTKs constitute a class of cell surface transmembrane proteins that play important roles in mediating extracellular to intracellular signaling. Humans carry approximately 60 RTKs (*Manning et al., 2002*), the *Drosophila* genome encodes 21 RTK genes (*Sopko and Perrimon, 2013*), *Bombyx mori* has 20 RTKs (*Alexandratos et al., 2016*), and the *German cockroach* genome identifies 16 RTKs (*Li et al., 2022*). *H. armigera* has 20 RTK candidates with gene codes in the *H. armigera* genome by our analysis. The cotton bollworm is a well-known and worldwide distributed agricultural pest in Lepidoptera, which threatens cotton and many other vegetable crops by rapidly producing resistance to various chemical insecticides and Bt-transgenic cotton. Using *H. armigera* as a model, we focus on identifying the RTKs functioning as the JH receptors and demonstrating the mechanism. We screened the RTKs sequentially, including examining the roles of 20 RTKs identified in the *H. armigera* genome in JH-regulated gene expression to obtain primary candidates, followed by screening of the candidates by their roles in maintaining larval status, JH-induced rapid increase of intracellular calcium levels, JH-induced phosphorylation of MET and TAI, and affinity to JH. The cadherin 96ca (CAD96CA) and fibroblast growth factor receptor 1 (FGFR1) were finally determined as JH cell membrane receptors by their roles in JH-regulated gene expression, maintaining larval status, JH-induced rapid increase of intracellular calcium levels, JH-induced phosphorylation of MET and TAI, and their JH-binding affinity. Their roles as JH cell membrane receptors were further determined by knockdown and knockout of them in vivo and cell lines, and overexpression of them in mammal HEK-293T heterogeneously. These data not only improve our knowledge of JH signaling and open the door to studying insect development but also present new targets to explore the new growth regulators to control the pest.

## Results

### The screen of the RTKs involved in JH signaling

To explore which RTKs may be involved in JH signaling, a total of RTKs were identified in the *H. armigera* genome. We found 20 RTK-like proteins encoded in the *H. armigera* genome and named the RTKs according to the nomenclature typically used in the genome or according to their homologues in *B. mori* or *D. melanogaster* (*Supplementary file 1*). Phylogenetic analysis showed that the 20 RTK candidates in *H. armigera* were conserved in *B. mori* and *D. melanogaster* (*Figure 1—figure supplement 1*). All the analyzed RTKs were grouped according to their structural characteristics and homology to the structure of 20 subfamilies of humans (*Honegger et al., 1989*; *Lemmon and Schlessinger, 2010*; *Sparrow et al., 1997*; *Yarden and Ullrich, 1988*); cell wall integrity and stress response component kinase (WSCK), tyrosine-protein kinase receptor torso like (TORSO) and serine/threonine-protein kinase STE20-like (STE 20-like) were not classified (*Figure 1—figure supplement 2*).

To identify the RTKs involved in JH III signaling, 20 RTKs of *H. armigera* were knocked down by RNA interference (RNAi) in HaEpi cells using JH III-induced *Kr-h1*, *Vg*, *Jhi-1*, and *Jhi-26* gene expression as readouts. When *Cad96ca*, *Drl* (encoding derailed), *Fgfr1*, *Nrk* (encoding neurotropic receptor kinase), *Vegfr1* (encoding vascular endothelial growth factor receptor 1), and *Wsck* were knocked down, respectively, JH III-upregulated expression of *Kr-h1* was decreased. However, knocking down

other *Rtks* did not decrease the *Kr-h1* transcription level. When *Cad96ca*, *Drl*, *Fgfr1*, *Nrk*, *Vegfr1*, *Wsck*, and *Inr* (encoding insulin-like receptor) were knocked down, JH III-upregulated expression of *Vg* was decreased. RNAi of *Rtks* did not affect JH-induced *Jhi-1* expression. When *Cad96ca*, *Fgfr1*, *Nrk*, and *Vegfr1* were knocked down, JH III-upregulated expression of *Jhi-26* was decreased (*Figure 1A*). *Rtks* were confirmed to be knocked down significantly in HaEpi cells (*Figure 1—figure supplement 3A*). The off-target effects of their knockdown were excluded from the genes we detected. Off-target genes were selected based on the identity rate of nucleotide sequences (*Figure 1—figure supplement 3B*). By the primary screening of RNAi, six RTKs, CAD96CA, DRL, FGFR1, NRK, VEGFR1, and WSCK were chosen for further screening.

The tissue-specific and developmental expression profiles of the six selected RTKs were determined using qRT-PCR to identify their possible roles in tissues at different developmental stages. The mRNA levels of *Vegfr1*, *Drl*, *Cad96ca*, and *Nrk* showed no tissue specificity in the epidermis, midgut, and fat body. Their transcript levels were high at the sixth instar feeding stage (6th–6 h to 6th–48 h) compared with those at the metamorphic molting stage (6th–72 h to 6th–120 h) and pupal stages (P–0 d to P–8 d). *Fgfr1* was highly expressed in the midgut at these feeding stages. *Wsck* was highly expressed from the 6th–48 h to the pupal stage and showed no tissue specificity (*Figure 1—figure supplement 4A*). These data suggested that the RTKs are differentially distributed in tissues and highly expressed during larval feeding stages.

We further examined the roles played by these six RTKs in JH III-delayed pupation by injecting double-stranded RNA (dsRNA) into the fifth instar 20 hr larval hemocoel. The interference of these six RTK genes in larvae led to a significant decrease in the expression of *Kr-h1*, in which, *Cad96ca*, *Nrk*, *Fgfr1*, and *Wsck* knockdown resulted in an increase in the expression of *Br-z7* (encoding broad isoform Z7) (*Figure 1—figure supplement 4B*). The pupation time was approximately 162 hr in 93% of the larvae in the dimethyl sulfoxide (DMSO) control group. After injection of JH III, the pupation time was approximately 187 hr in 76% of the larvae, which was 25 hr later than that of the DMSO control group, suggesting that JH III delayed pupation. In the *dsGFP* +JH III-injected control, 70% of larvae pupated at approximately the same time as larvae after JH III treatment. In the *dsVegfr1* +JH III and *dsDrl* +JH III treatment groups, 60–63% of larvae exhibited delayed pupation; only 9–10% of the larvae did not show delayed pupation, and 28–30% died at the larval or pupal stage. However, 66–68% of the larvae did not show delayed pupation after *dsCad96ca* +JH III, *dsNrk* +JH III, *dsFgfr1* +JH III or *dsWsck* +JH III injection (*Figure 1B and C* and *Figure 1—figure supplement 4C*). These results indicated that CAD96CA, NRK, FGFR1, and WSCK are involved in JH III-induced delayed pupation.

To address the mechanism of the RTK effects on JH signaling, we examined the roles played by the selected RTKs in JH III-induced cellular responses by knocking down RTK gene expression in HaEpi cells. JH III-induced rapid calcium mobilization was repressed after knockdown of *Vegfr1*, *Drl*, *Cad96ca*, *Nrk*, *Fgfr1*, or *Wsck* compared with that after *dsGFP* knockdown (*Figure 2A*). The efficacy of RNAi was confirmed (*Figure 2B*). However, only the knockdown of *Cad96ca*, *Nrk*, and *Fgfr1* decreased the JH III-induced phosphorylation of MET1 and TAI (*Figure 2C*). The results suggested that these RTKs are involved in JH III-induced rapid cellular calcium increase but are differentially involved in JH III-induced MET1 and TAI phosphorylation.

## CAD96CA and FGFR1 had high affinity to JH III

Since Cad96CA, FGFR1, and NRK were not only involved in JH-regulated *Kr-h1* expression, JH III-induced delayed pupation, and calcium levels increase, but also involved in MET and TAI phosphorylation, we further analyzed their binding affinity to JH III. OTK did not respond to JH III, so we used it as a control protein on the cell membrane to exclude the possibility of nonspecific binding. The affinity of CAD96CA, FGFR1, NRK, and OTK for JH III was determined using saturable specific–binding curve analysis via microscale thermophoresis (MST). The experiment used full–length sequences of CAD96CA, FGFR1, NRK, and OTK. CAD96CA-CopGFP-His (CopGFP is a 26 kDa green fluorescent protein cloned from copepod *Pontellina plumate*), FGFR1-CopGFP-His, NRK-CopGFP-His, and OTK-CopGFP-His were overexpressed in the Sf9 cell line (Sf9 cells expressed the proteins at a higher level than HaEpi cells) and then, the proteins were isolated separately to determine the JH III-binding strength of each. Immunocytochemistry showed that CAD96CA-CopGFP-His, FGFR1-CopGFP-His, NRK-CopGFP-His, and OTK-CopGFP-His located in the plasma membrane (*Figure 3A*). The purity of the proteins was assessed and confirmed using sodium dodecyl sulfate-polyacrylamide

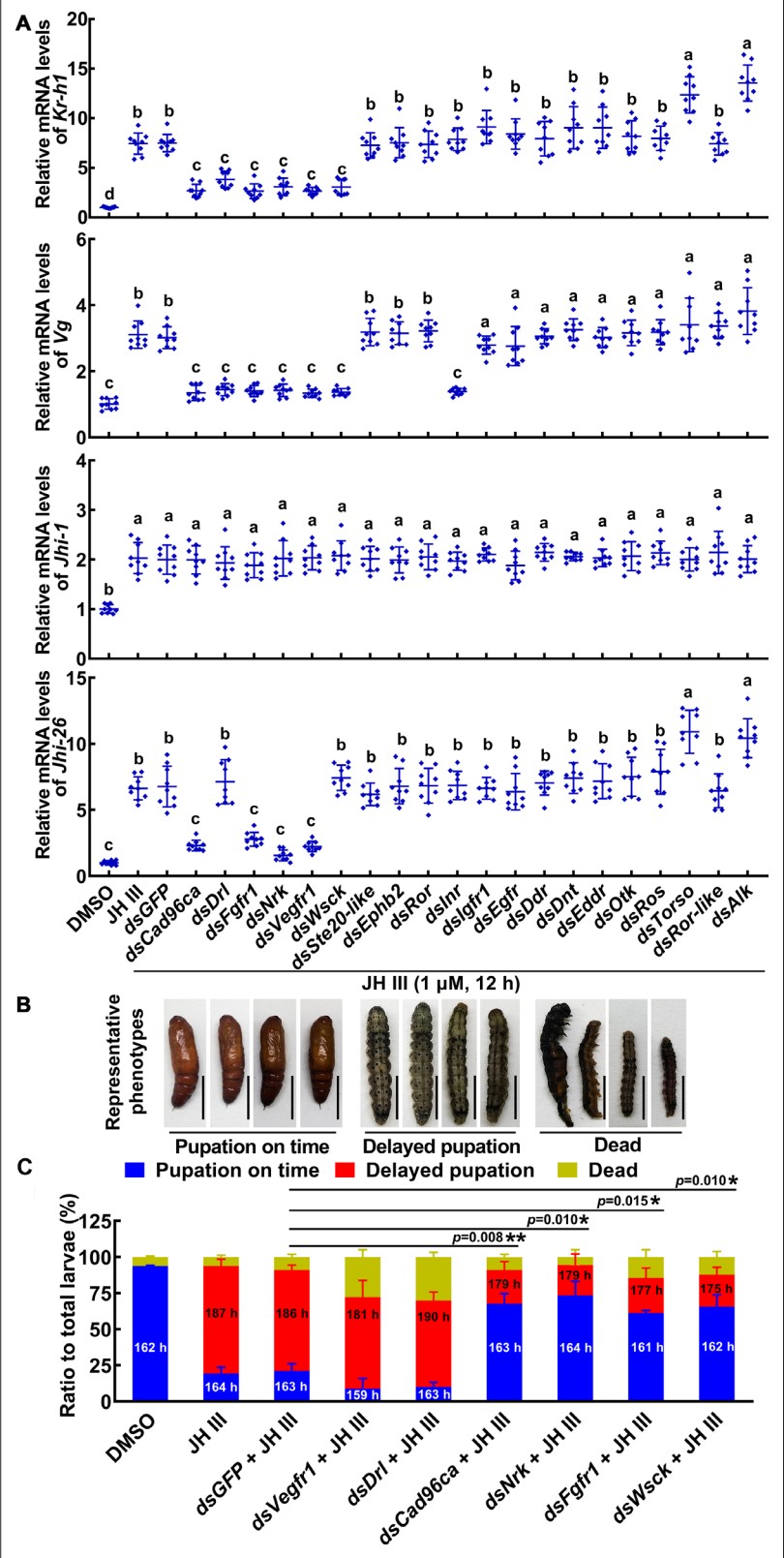

**Figure 1.** RTKs were screened to determine their involvement in the JH signal pathway in HaEpi cells and larvae.
(**A**) The roles of RTKs in JH III-induced *Kr-h1*, *Vg*, *Jhi-1*, and *Jhi-26* expression were determined by RNAi of *Rtk* genes (1 μg/mL dsRNA, 48 hr, 1 μM JH III for 12 hr). DMSO as solvent control. The relative mRNA levels were calculated via the $2^{-\Delta\Delta CT}$ method and the bars indicate the mean ± SD. n=3. Multiple sets of data were compared

*Figure 1 continued on next page*

*Figure 1 continued*

by ANOVA. The different lowercase letters show significant differences. (**B**) The examples of phenotype after *Vegfr1*, *Drl*, *Cad96ca*, *Nrk*, *Fgfr1*, and *Wsck* knockdown in larvae. Scale = 1 cm. (**C**) Phenotype percentage and pupation time after *Vegfr1*, *Drl*, *Cad96ca*, *Nrk*, *Fgfr1*, and *Wsck* knockdown in larvae. The time was recorded from the bursting of the head shell of the 5th instar to pupal development. Images were collected after more than 80% of the larvae had pupated in the DMSO control group. Two-group significant differences were calculated using Student's *t* test (*p<0.05, **p<0.01 indicate the significant difference between the percentages of the delayed pupation in *dsGFP* +JH III control group and gene knockdown) based on three replicates, n=30 × 3 larvae.

The online version of this article includes the following source data and figure supplement(s) for figure 1:

**Source data 1.** Statistical data for *Figure 1A and C*.

**Figure supplement 1.** Phylogenetic tree analysis to identify RTKs of *H. armigera*.

**Figure supplement 2.** Structural characteristics of the RTK domains.

**Figure supplement 3.** The interference efficiency of dsRNA and off−target detection.

**Figure supplement 3—source data 1.** Statistical data for *Figure 1—figure supplement 3*.

**Figure supplement 4.** Expression profiles, interference efficiency and phenotype of 6 *Rtks* in larvae.

**Figure supplement 4—source data 1.** Statistical data for *Figure 1—figure supplement 4A and B*.

gel electrophoresis (SDS–PAGE) with Coomassie brilliant blue staining (*Figure 3B*). CAD96CA-CopGFP-His binding to JH III exhibited a dissociation constant (Kd)=11.96 ± 1.61 nM. Similarly, the saturable specific binding of FGFR1-CopGFP-His to JH III exhibited a Kd = 23.61 ± 0.90 nM, and NRK-CopGFP-His and OTK-CopGFP-His showed no obvious binding (*Figure 3C*). These results suggested that CAD96CA and FGFR1 bind JH III.

The JH intracellular receptor MET has been reported to bind to JH in *Tribolium* (*Charles et al., 2011*); therefore, MET1 in *H. armigera* was used as the positive control in analyses to assess the applicability of the MST method. MET1-CopGFP-His and CopGFP-His were overexpressed in the Sf9 cell line and then isolated to determine their binding strength to JH III. Immunocytochemistry showed the nuclear location of MET1 (*Figure 3—figure supplement 1A*). The purities of the isolated CopGFP-His and MET1-CopGFP-His proteins were examined and confirmed using SDS–PAGE with coomassie brilliant blue staining (*Figure 3—figure supplement 1B*). The saturable specific binding of MET1-CopGFP-His to JH III exhibited a Kd = 6.38 ± 1.41 nM. CopGFP-His showed weaker binding to JH III (*Figure 3—figure supplement 1C*). In comparison with the Kd of *Tribolium* MET to JH III of 2.94±0.68 nM as detected by [$^3$H]JH III (*Charles et al., 2011*), the MST method was a valid approach to detect the JH III binding to a protein.

To further validate CAD96CA and FGFR1 binding JH III, saturation assays were performed using the analogs of JH, the farnesol, methoprene, and farnesoate (MF). Results showed that CAD96CA-CopGFP-His bound farnesol with a Kd of 1039.2±0.68 nM. CAD96CA-CopGFP-His bound metho-prene with a Kd of 553.94±1.11 nM. CAD96CA-CopGFP-His bound methyl farnesoate (MF) with a Kd of 446.55±0.80 nM. CAD96CA-CopGFP-His bound JH III with a Kd of 12.10±1.4 nM (*Figure 3D*). The results confirmed that CAD96CA has the highest affinity to JH III.

Because methoprene is known as an effective juvenoid (*Konopova and Jindra, 2007*) and competes with JH III in binding to MET (*Charles et al., 2011*), the competitive experiment was performed to confirm CAD96CA bound JH III. CAD96CA-CopGFP-His bound to methoprene plus JH III with a Kd value of 261.43±0.81 nM, whereas CAD96CA-CopGFP-His bound to methoprene with a Kd value of 563.49±0.7 nM (*Figure 3E*). These suggested that CAD96CA-CopGFP-His has the highest affinity to JH III compared with the analogs.

Similarly, the saturable specific binding of FGFR1-CopGFP-His bound farnesol with a Kd = 23810 ± 0.51 nM; FGFR1-CopGFP-His bound methoprene with a Kd = 529.68 ± 0.60 nM; FGFR1-CopGFP-His bound to MF exhibited a Kd = 417.20 ± 0.66 nM; and FGFR1-CopGFP-His bound to JH III exhibited a Kd = 21.45 ± 1.02 nM (*Figure 3F*), suggesting FGFR1 had the highest affinity to JH III. The competitive binding of FGFR1-CopGFP-His to methoprene plus JH III with a Kd value = 349.27 ± 0.58 nM, whereas FGFR1-CopGFP-His bound to methoprene with a Kd value 523.57±0.89 nM (*Figure 3G*). These suggested that FGFR1 has the highest affinity to JH III compared with the analogs.

Various mutants of CAD96CA and FGFR1 were further constructed to identify the key motifs in CAD96CA and FGFR1 critical for JH binding. Truncated mutations were performed on extracellular

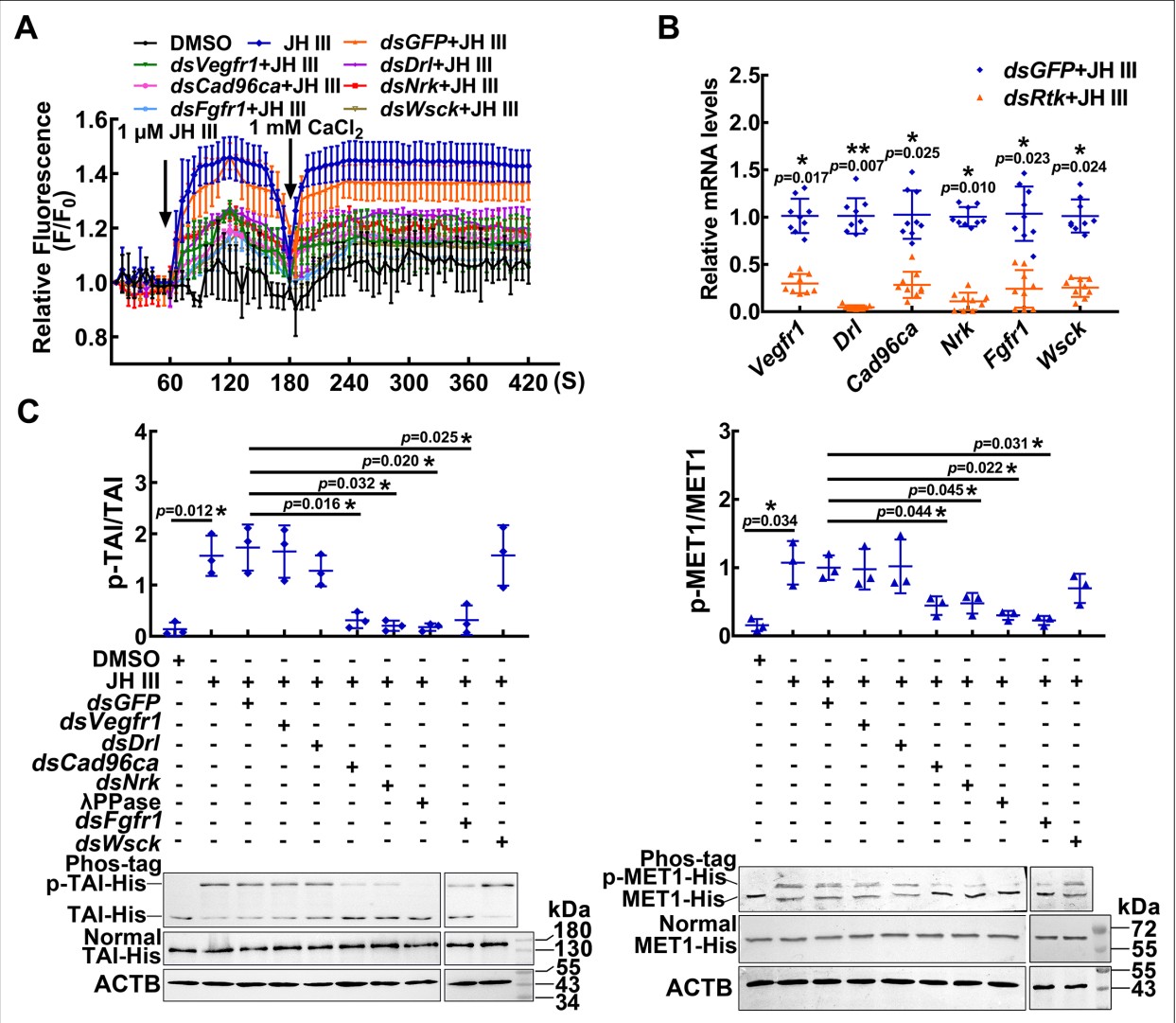

**Figure 2.** RTKs involved in JH III-regulated $Ca^{2+}$ increase and protein phosphorylation. (**A**) The level of $Ca^{2+}$ after *Vegfr1*, *Drl*, *Cad96ca*, *Nrk*, *Fgfr1*, and *Wsck* knockdown in HaEpi cells. The cells were incubated with dsRNA (the final concentration was 1 µg/mL for 48 hr) and AM ester calcium crimson dye (3 µM, 30 min). $F_0$: the fluorescence intensity of HaEpi cells without treatment. F: the fluorescence intensity of HaEpi cells after different treatments. DMSO as solvent control. (**B**) The interference efficiency of dsRNA in HaEpi cells. (**C**) Western blotting was performed to analyze TAI-His and MET1-His phosphorylation after treatment with dsRNA and JH III (1 µM, 3 hr). Phos-tag: phosphate affinity SDS−PAGE gel, Normal: normal SDS−PAGE gel, which was a 7.5 or 10% SDS−PAGE gel. The results of three independent repeated western blots were statistically analyzed by ImageJ software. The *p* value was calculated by Student's *t* test based on three independent replicate experiments, n=3. The error bar indicates the mean ± SD.

The online version of this article includes the following source data for figure 2:

**Source data 1.** Statistical data for *Figure 2A, B and C*.

**Source data 2.** PDF file containing original western blots for *Figure 2C*, indicating the relevant bands and treatments.

**Source data 3.** Original files for western blot analysis displayed in *Figure 2C*.

regions of CAD96CA and FGFR1, including CAD96CA-M1 (51–615 AA, amino acid), CAD96CA-M2 (101–615 AA), CAD96CA-M3 (151–615 AA), CAD96CA-M4 (201–615 AA), FGFR1-M1 (101–615 AA), FGFR1-M2 (201–615 AA), FGFR1-M3 (301–615 AA), and FGFR1-M4 (401–615 AA). Mutants were overexpressed, and the encoded mutants located in the plasma membrane, as confirmed via immunocytochemistry, and the purity of the proteins was confirmed using SDS−PAGE with coomassie brilliant blue staining (*Figure 3—figure supplement 1D–I*). Compared with the wild-type counterparts, CAD96CA-M2, CAD96CA-M3, and CAD96CA-M4 mutants' affinity to JH III was significantly reduced (*Figure 3H*). Similarly, FGFR1-M2, FGFR1-M3, and FGFR1-M4 mutants for the binding affinity to JH III

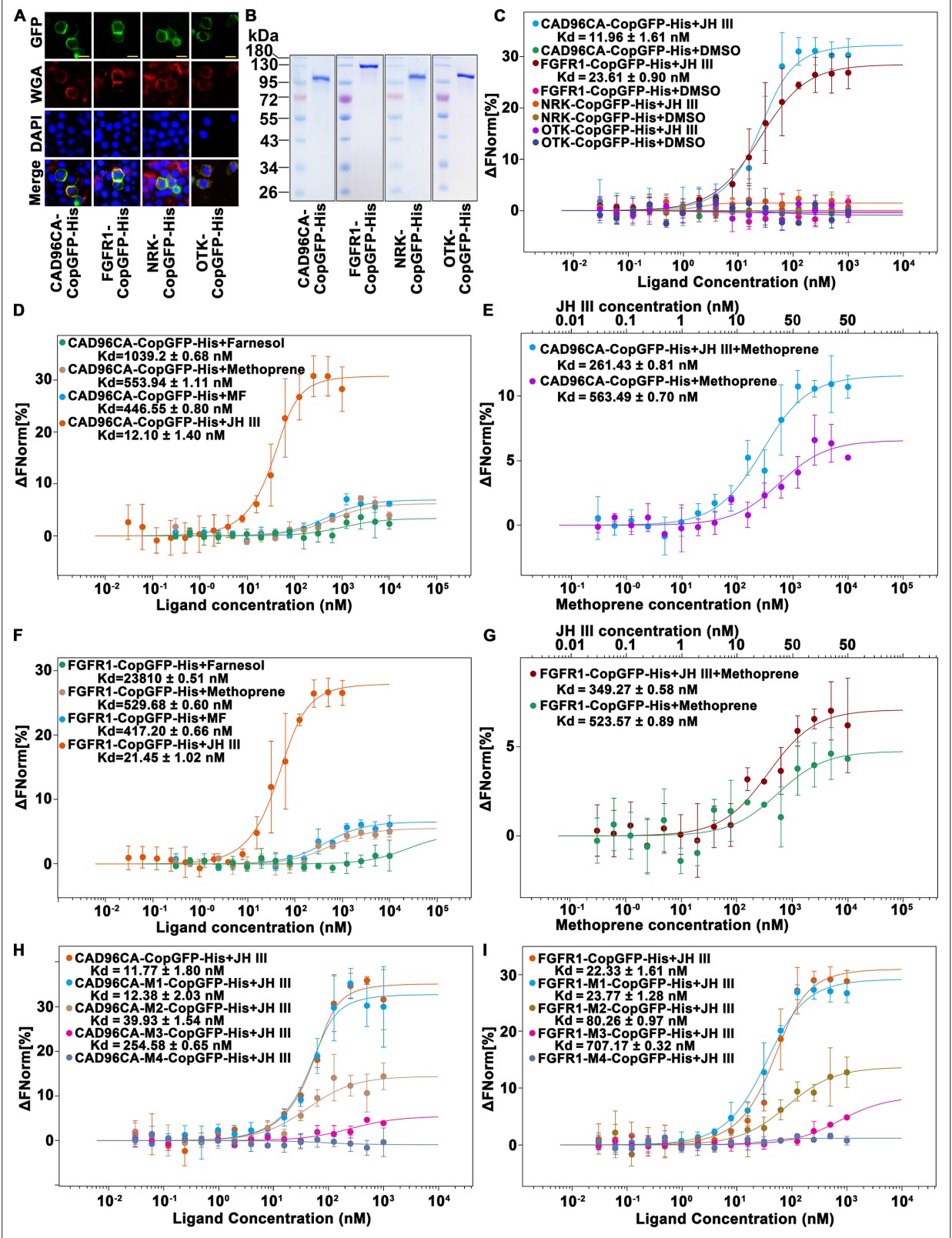

**Figure 3.** CAD96CA and FGFR1 could bind JH III. (**A**) Cell membrane localization of the overexpressed CAD96CA-CopGFP-His, FGFR1-CopGFP-His, NRK-CopGFP-His and OTK-CopGFP-His. GFP: green fluorescence of RTKs fused with a green fluorescent protein. WGA: red fluorescence, the cell membrane was labeled with wheat germ agglutinin. DAPI: nuclear staining. Merge: the pictures of different fluorescent-labeled cells were combined. The cells were observed with a fluorescence microscope. Scale bar = 20 μm. (**B**) Coomassie brilliant blue staining of the SDS−PAGE gel showed the

*Figure 3 continued*

purity of the separated CAD96CA-CopGFP-His, FGFR1-CopGFP-His, NRK-CopGFP-His, and OTK-CopGFP-His proteins. (**C**) Saturation binding curves of CAD96CA-CopGFP-His, FGFR1-CopGFP-His, NRK-CopGFP-His and OTK-CopGFP-His. (**D**) Saturation binding curves of CAD96CA-CopGFP-His were incubated with the indicated compounds. (**E**) The binding and competition curves of CAD96CA and methoprene. (**F**) Saturation binding curves of FGFR1-CopGFP-His were incubated with the indicated compounds. (**G**) The binding and competition curves of FGFR1 and methoprene. (**H**) The binding curves of CAD96CA mutants and JH III. (**I**) The binding curves of FGFR1 mutants with JH III. Data are mean ± SE of three replicates. n=3.

The online version of this article includes the following source data and figure supplement(s) for figure 3:

**Source data 1.** Statistical data for *Figure 3C-I*.

**Source data 2.** PDF file containing original gels for *Figure 3B*, indicating the relevant bands and treatments.

**Source data 3.** Original files for gel analysis displayed in *Figure 3B*.

**Figure supplement 1.** MET1 bound JH III, and CAD96CA and FGFR1 mutants.

**Figure supplement 1—source data 1.** Statistical data for *Figure 3—figure supplement 1C*.

**Figure supplement 1—source data 2.** PDF file containing original gels for *Figure 3—figure supplement 1B, F and I*, indicating the relevant bands and treatments.

**Figure supplement 1—source data 3.** Original files for gel analysis displayed in *Figure 3—figure supplement 1B, F and I*.

**Figure supplement 2.** CAD96CA and FGFR1 bound JH III were analyzed using ITC.

**Figure supplement 2—source data 1.** Statistical data for *Figure 3—figure supplement 2*.

---

were significantly reduced (*Figure 3I*). These results suggested that the extracellular domain 51–151 AA in CAD96CA and the extracellular domain 101–301 AA in FGFR1 play a vital role in JH binding.

The affinity of CAD96CA, FGFR1, NRK, and OTK for JH III was further determined using saturable specific–binding curve analysis via isothermal titration calorimetry (ITC). ITC was used as an alternative method to further examine the affinity of CAD96CA and FGFR1 to JH III. CAD96CA-CopGFP-His bound JH III with a Kd value of 79.6±27.5 nM. Similarly, the saturable specific binding of FGFR1-CopGFP-His to JH III with a Kd value of 88.5±19.4 nM, and NRK-CopGFP-His and OTK-CopGFP-His showed no remarkable binding (*Figure 3—figure supplement 2*). These results also suggested that CAD96CA and FGFR1 bind JH III.

## Gene knockout of *Cad96ca* or *Fgfr1* by CRISPR/Cas9 caused early pupation and a decrease of JH signaling

To verify the roles of CAD96CA and FGFR1 in JH signaling in vivo, we mutated *Cad96ca* or *Fgfr1* by CRISPR/Cas9 technology, individually. We synthesized two gRNAs targeting different sites in the *Cad96ca* and *Fgfr1* coding regions with a low probability of causing off–target effects. Two gRNAs (referred to as *Cad96ca*-gRNAs) are located at the third exon of the *Cad96ca* gene (*Figure 4A*), and two gRNAs (referred to as *Fgfr1*-gRNAs), which are located at the second exon of the *Fgfr1* gene (*Figure 4B*) were used for the experiment.

When the Cas9-gRNA injected eggs (105 eggs were injected each for three injections, a total of 315 experimental eggs) had developed into second instar larvae, the survival rates were determined. The survival rate of the Cas9-gRNA-injected eggs (19.4~20.6%) did not significantly differ from that of the control eggs injected with Dulbecco's phosphate-buffered saline (DPBS; a survival rate of 22.6%), suggesting that the mixture of gRNA and Cas9 protein was nontoxic to the *H. armigera* eggs. In 61 survivors of Cas9 protein plus *Cad96ca*-gRNA injection, 30 mutants were identified, and mutation efficiency was 49.2%. Similarly, in the 65 survivors of Cas9 protein plus *Fgfr1*-gRNA injection, 35 mutants were identified, with a mutation efficiency of 53.8% (*Figure 4C*). The DNA sequences, deduced amino acids, and off-target were analyzed (*Figure 4—figure supplement 1*). Most wild-type larvae showed a phenotype of pupation on time. However, in the *Cad96ca* mutants, 86% of the larvae showed early pupation, with the pupation time being 24 hr earlier than the control. In the *Fgfr1* mutants, 91% of the larvae showed early pupation, with the pupation time being 23 hr earlier than the control (*Figure 4D and E*). The data suggested that both CAD96CA and FGFR1 prevent pupation in vivo, which is consistent with the role of the JH preventing pupation.

To address the mechanism of early pupation caused by the knockout of *Cad96ca* or *Fgfr1*, we compared the expression of the genes in the JH and 20E pathways between mutant and wild-type *H. armigera*. Both the mutants of *Cad96ca* and *Fgfr1* led to a significant decrease in *Kr-h1*, a JH-induced

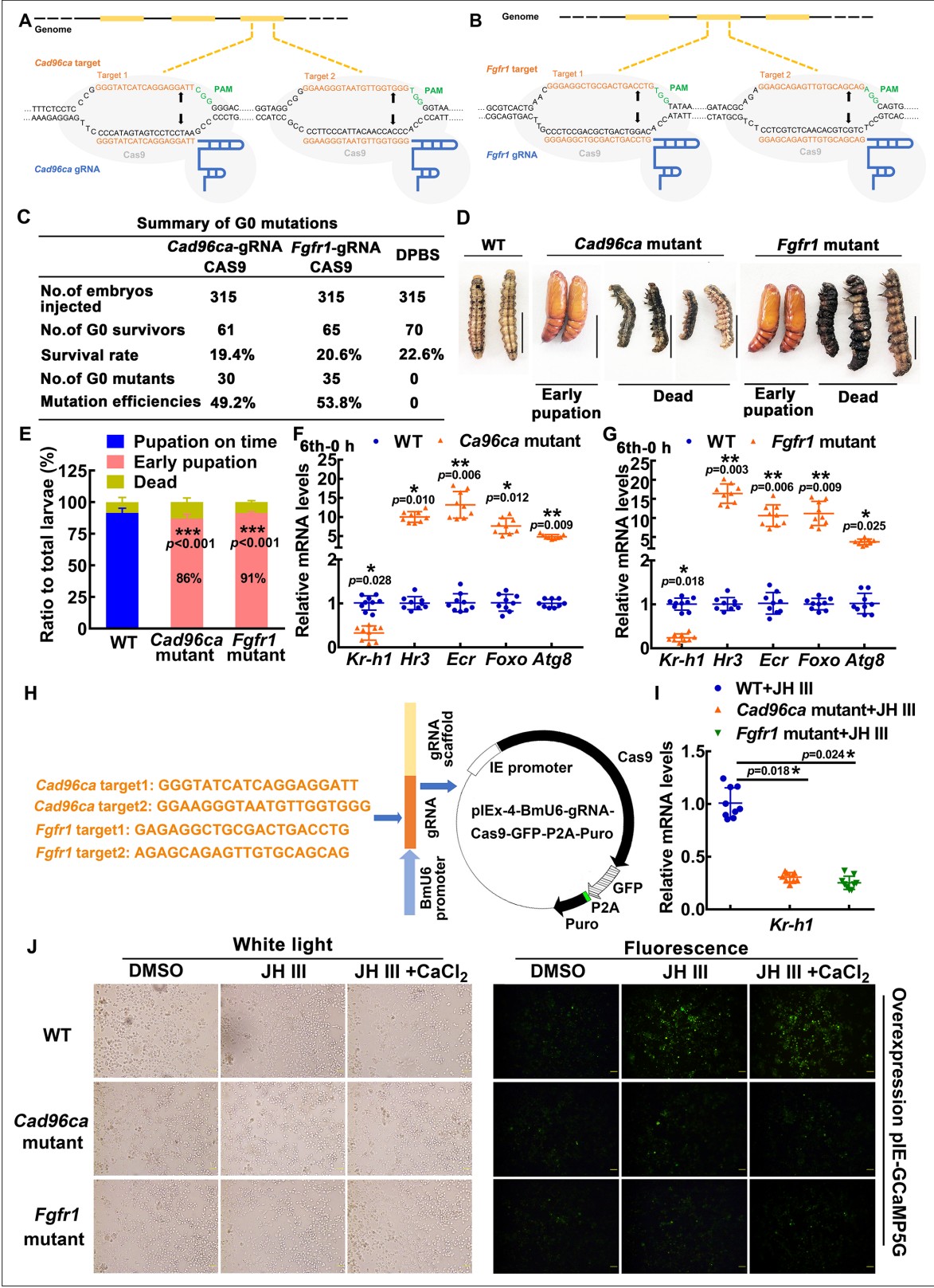

**Figure 4.** The roles of CAD96CA and FGFR1 in larval development were determined by CRISPR/Cas9 system-mediated mutants. (**A, B**) Schematic showing the injection mixture of the CRISPR/Cas9 system. The black line refers to the genome of *H. armigera*; the yellow blocks correspond to exons. The Cas9 nuclease (in grey) was targeted to genomic DNA by *Cad96ca*-gRNA or *Fgfr1*-gRNA with an ~20 nt guide sequence (orange) and a scaffold (blue). The guide sequence pairs with the DNA target (orange sequence on the top strand), which requires the upstream sequence of the 5'-CGG-3'

*Figure 4 continued on next page*

*Figure 4 continued*

adjacent motif (PAM; green). Cas9 induces a double-strand break (DSB) ~3 bp upstream of the PAM (black triangle). (**C**) Summary of G0 mutations. (**D**) Images showing WT and mutant *H. armigera* phenotypes. The scale represents 1 cm. (**E**) Morphology and statistical analysis of WT and mutant *H. armigera*. Both *Cad96ca* and *Fgfr1* mutant larvae showed earlier pupation than WT controls. (**F, G**) qRT−PCR showing the mRNA levels of the JH/20E response genes in WT and mutant *H. armigera*. (**H**) Schematic showing the CRISPR/Cas9 editing in HaEpi cells by pIEx-4-BmU6-*Cad96ca*-gRNA-Cas9-GFP-P2A-Puro and pIEx-4-BmU6-*Fgfr1*-gRNA-Cas9-GFP-P2A-Puro recombination vectors. (**I**) qRT−PCR showing the mRNA levels of *Kr-h1* in WT and mutant HaEpi cells. (**J**) pIEx-GCaMP5G was overexpressed in WT and mutant HaEpi cells, and calcium mobilization was detected. Green fluorescence shows the calcium signal. The concentration of JH III was 1 μM, and that of CaCl₂ was 1 mM. The scale bar represents 100 μm.

The online version of this article includes the following source data and figure supplement(s) for figure 4:

**Source data 1.** Statistical data for *Figure 4E, F, G, and I*.

**Figure supplement 1.** Targeted mutagenesis of *Cad96ca* and *Fgfr1* in *H. armigera*.

**Figure supplement 2.** Targeted mutagenesis of *Cad96ca* and *Fgfr1* in HaEpi cells.

**Figure supplement 2—source data 1.** Statistical data for *Figure 4—figure supplement 2E*.

gene functioning at the downstream of MET1, expression and an increase in 20E pathway gene expression, including *Hr3*, *Ecr*, *Foxo* and *Atg8* compared with the wild-type *H. armigera*, respectively (**Figure 4F and G**), indicating that CAD96CA and FGFR1 prevented pupation by increasing *Kr-h1* expression and repressing 20E pathway gene expression, which suggests CAD96CA and FGFR1 play roles in JH signaling.

To confirm the roles played by CAD96CA and FGFR1 in JH signaling, we further examined the response of HaEpi cells to JH III induction after editing of *Cad96ca* and *Fgfr1* by CRISPR/Cas9 in HaEpi cells using the gRNAs inserted into the pIEx-4-BmU6-gRNA-Cas9-GFP-P2A-Puro plasmid (**Figure 4H**). The mutation of *Cad96ca* and *Fgfr1* in HaEpi cells was confirmed by sequencing the mutants and deduced amino acids (**Figure 4—figure supplement 2A–D**). *Cad96ca* or *Fgfr1* mutation repressed the JH III-induced expression of *Kr-h1* in HaEpi cells compared with wild-type cells (**Figure 4I**), and repressed the JH III-induced rapid calcium mobilization in cells (**Figure 4J** and **Figure 4—figure supplement 2E**), suggesting that CAD96CA and FGFR1 were involved in JH III-induced expression of *Kr-h1* and rapid calcium mobilization. These results supported the hypothesized roles played by CAD96CA and FGFR1 in JH signaling.

To clarify the physiological roles and relationships of intracellular receptor MET1, cell membrane receptors CAD96CA and FGFR1 in JH signaling, we knocked out *Met1* individually (single knockout), *Cad96ca,* and *Fgfr1* together (double knockout), and *Met1*, *Cad96ca* and *Fgfr1* together (triple knockout) using CRISPR/Cas9. Sequencing and genotypic analysis for mutant identification (**Figure 5A** and **Figure 5—figure supplement 1**). In the hatched larvae, the mutation rates were 25% for *Met1*, 21% for *Cad96ca* and *Fgfr1*, and 17% for *Met1*, *Cad96ca* and *Fgfr1* mutations, respectively (**Figure 5B**). In the G0 generation larvae, 86% of the wild-type individuals pupated normally, with an average pupation time of approximately 20 days. In *Met1* mutants, 42% of the larvae pupated prematurely, with an average pupation time of about 16 days, and 58% of the individuals died during the transition from larvae to pupae. In *Cad96ca* and *Fgfr1* double mutants, 70% of the individuals pupated prematurely, with an average pupation time of around 17 days, and 30% of the individuals died during the transition from larvae to pupae. In *Met1*, *Cad96ca* and *Fgfr1* triple mutants, 19% of the individuals pupated prematurely, with an average pupation time of approximately 15 days, and 81% of the individuals died at the 5th instar and 6th instar larval stages. Among the surviving pupae, in *Met1* mutants, 84% died and 16% of the pupae developed into abnormal adults. In *Cad96ca* and *Fgfr1* double mutants, 72% of the pupae died and 28% of the pupae developed into abnormal adults. In *Met1*, *Cad96ca* and *Fgfr1* triple mutants, 92% of the pupae died, and 8% of the pupae developed into abnormal adults (**Figure 5C and D**). The expression level of *Kr-h1* was significantly decreased in these mutants, while the expression levels of genes related to the 20E pathway were significantly increased (**Figure 5E**). These results indicated that CAD96CA and FGFR1 play important roles in the JH signaling, suggesting that MET1, CAD96CA, and FGFR1 can transmit JH signal to prevent pupation independently and cooperatively.

## CAD96CA and FGFR1 transmitted JH signal in different insect cells and HEK-293T cells

To demonstrate the universality of CAD96CA and FGFR1 in JH signaling in different insect cells, we investigated JH-triggered calcium ion mobilization and *Kr-h1* expression in Sf9 cells developed

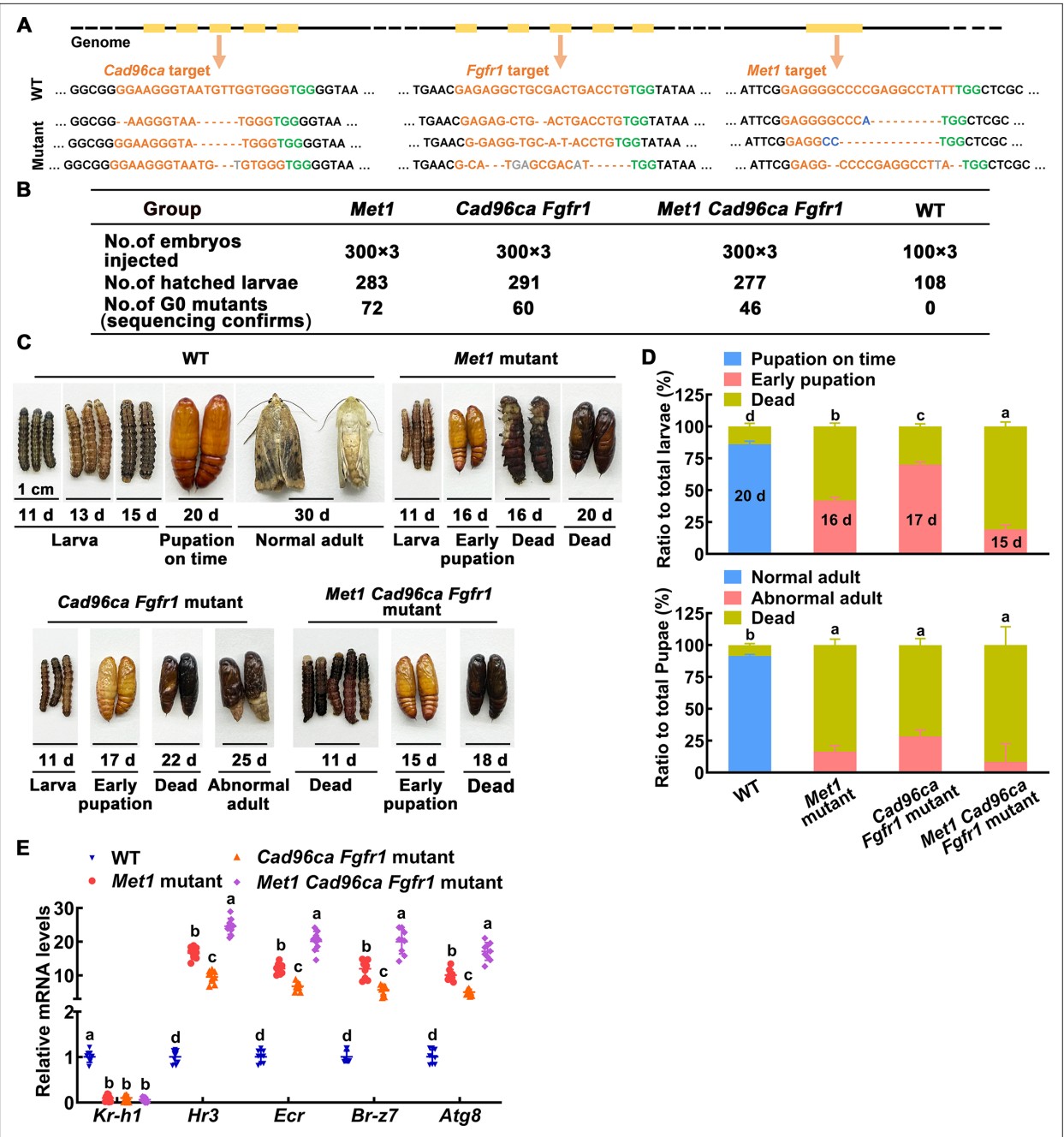

**Figure 5.** The roles of MET1, CAD96CA, and FGFR1 in larval development were determined by CRISPR/Cas9 system-mediated mutants. (**A**) Various types of mutations caused by CRISPR/Cas9 system. Deletions were indicated by hyphens; insertions were shown by gray letters; substitutions were shown in blue letters. (**B**) The number of individuals in the G0 generation of the three experiments. (**C**) Images showing WT and mutant *H. armigera* phenotypes. The scale represents 1 cm. (**D**) Morphology and statistical analysis of WT and mutant *H. armigera*. (**E**) qRT−PCR showing the mRNA levels of the JH/20E response genes in WT and mutant *H. armigera*.

The online version of this article includes the following source data and figure supplement(s) for figure 5:

**Source data 1.** Statistical data for *Figure 5D and E*.

**Figure supplement 1.** Targeted mutagenesis of *Cad96ca*, *Fgfr1*, and *Met1* in *H. armigera*.

from *S. frugiperda* and S2 cells developed from *D. melanogaster*. JH III induced intracellular Ca²⁺ release and extracellular Ca²⁺ influx in Sf9 and S2 cells, but DMSO could not. However, knockdown of *Cad96ca* and *Fgfr1* (named *Htl* or *Btl* in *D. melanogaster*), respectively, significantly decreased JH III-induced intracellular Ca²⁺ release and extracellular Ca²⁺ influx (*Figure 6A and B*), and *Kr-h1* expression

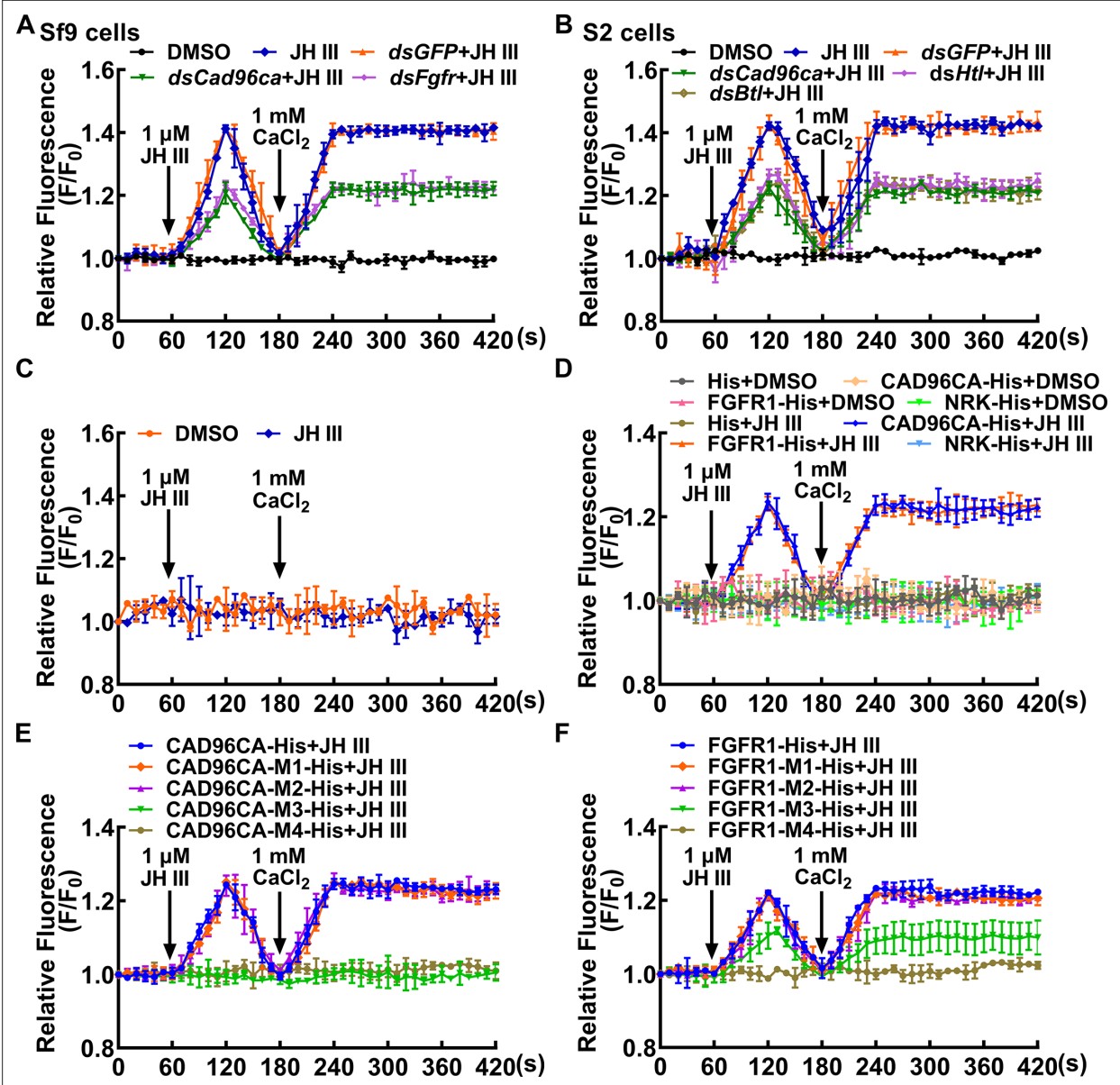

**Figure 6.** CAD96CA and FGFR1 participated in JH-induced calcium ion mobilization. (**A**) The level of $Ca^{2+}$ after *Cad96ca* and *Fgfr* knockdown in Sf9 cells. The cells were incubated with dsRNA (the final concentration was 1 µg/mL for 48 hr). $F_0$: the fluorescence intensity of Sf9 cells without treatment. F: the fluorescence intensity of Sf9 cells after different treatments. DMSO as solvent control. (**B**) Effect of JH III on calcium ion levels in S2 cells after *Cad96ca* and *Htl* knockdown. (**C**) The response of calcium ion levels to JH III in HEK-293T cells. (**D**) The analysis of calcium ion flow after HEK-293T cells overexpressed RTK. DMSO as solvent control. His as tag control. (**E, F**) The calcium was quantitated after HEK-293T cells overexpressed CAD96CA-His, FGFR1-His, and mutants. The error bar indicates the mean ± SD. n=3.

The online version of this article includes the following source data and figure supplement(s) for figure 6:

**Source data 1.** Statistical data for *Figure 6A-F*.

**Figure supplement 1.** The gene expression was analyzed by qPCR.

**Figure supplement 1—source data 1.** Statistical data for *Figure 6—figure supplement 1*.

**Figure supplement 2.** CAD96CA, FGFR1 and mutants overexpressed in HEK-293T cells.

**Figure supplement 2—source data 1.** PDF file containing original western blots for *Figure 6—figure supplement 2*, indicating the relevant bands and treatments.

**Figure supplement 2—source data 2.** Original files for western blot analysis displayed in *Figure 6—figure supplement 2*.

(*Figure 6—figure supplement 1A and B*). The efficacy of RNAi of *Cad96ca* and *Fgfr1* was confirmed in the cells (*Figure 6—figure supplement 1C and D*), suggesting that CAD96CA and FGFR1 had a general function to transmit JH signal in *S. frugiperda* and *D. melanogaster*.

To confirm the roles of CAD96CA and FGFR1 in transmitting JH signal, CAD96CA and FGFR1 of *H. armigera* were overexpressed heterogeneously in mammalian HEK-293T cells to exclude the unknown endogenous effect in insect cells. Immunocytochemistry showed that CAD96CA-GFP, FGFR1-GFP, and NRK-GFP locate in the plasma membrane. The proteins were confirmed using western blotting (*Figure 6—figure supplement 2A*). The wild HEK-293T cells had no significant changes in calcium ion levels under JH III induction, because there were no CAD96CA and FGFR1 in the cells (*Figure 6C*), indicating that HEK-293T cells did not respond to JH III induction. However, when HEK-293T cells overexpressed CAD96CA or FGFR1, respectively, JH III triggered rapid cytosolic $Ca^{2+}$ release and influx, by comparison with the DMSO condition, His tag, and other RTK NRK-His controls (*Figure 6D*). These results further confirmed that CAD96CA and FGFR1 transmit JH III signal.

CAD96CA and FGFR1 mutants were used to further confirm their roles in transmitting the JH signal in human HEK-293T cells. Mutants were overexpressed, and the encoded mutants located in the plasma membrane, as confirmed via immunocytochemistry, and the proteins were confirmed using western blotting (*Figure 6—figure supplement 2B*). Results showed that $Ca^{2+}$ increase was not detected in mutants CAD96CA-M3 and CAD96CA-M4 under JH III induction (*Figure 6E*), JH III-induced $Ca^{2+}$ mobilization was slightly detected in FGFR1-M3, and it was not detected in FGFR1-M4 (*Figure 6F*). These results confirmed that CAD96CA and FGFR1 play roles in transmitting JH III signal.

## Discussion

JH regulates insect development through intracellular and membrane signaling; however, the cell membrane receptors and the mechanism are unclear. In this study, CAD96CA and FGFR1 were screened out from the total 20 RTKs in the *H. armigera* genome and identified as JH III cell membrane receptors, which transmit JH signal for gene expression to prevent pupation and have a high affinity to JH III.

### CAD96CA and FGFR1 transmit JH signal

JH induces a set of gene expression, such as *Kr-h1* (*Truman, 2019*), *Vg* (*Roy et al., 2018*; *Song et al., 2014*), *Jhi-1*, and *Jhi-26* (*Dubrovsky et al., 2000*), a rapid intracellular calcium increase, phosphorylation of MET and TAI (*Liu et al., 2015*), and prevents pupation. We found that several RTKs are involved in JH III-induced gene expression and calcium increase; however, only *Cad96ca*, *Nrk*, *Fgfr1*, and *Wsck* are involved in the JH III-induced pupation delay, in which, only CAD96CA, NRK, and FGFR1 are involved in the JH-induced phosphorylation of MET1 and TAI, and only CAD96CA and FGFR1 can bind JH III. Therefore, CAD96CA and FGFR1 are finally determined as JH III receptors.

CAD96CA (also known as Stitcher, Ret-like receptor tyrosine kinase) activates upon epidermal wounding in *Drosophila* (*Tsarouhas et al., 2014*) and promotes growth and suppresses autophagy in the *Drosophila* epithelial imaginal wing discs (*O'Farrell et al., 2013*). There is a CAD96CA in the genome of the *H. armigera*, which is without function study. Here, we reported that CAD96CA prevents pupation by transmitting JH signal as a JH cell membrane receptor. We also showed that CAD96CA of other insects has a universal function of transmitting JH signal to trigger $Ca^{2+}$ mobilization, as demonstrated by the study in Sf9 cell lines of *S. frugiperda* and S2 cell lines of *D. melanogaster*.

FGFRs control cell migration and differentiation in the developing embryo of *D. melanogaster* (*Muha and Müller, 2013*). The ligand of FGFR is FGF in *D. melanogaster* (*Du et al., 2018*). FGF binds FGFR and triggers cell proliferation, differentiation, migration, and survival (*Beenken and Mohammadi, 2009*; *Lemmon and Schlessinger, 2010*). Three FGF ligands and two FGF receptors (FGFRs) are identified in *Drosophila* (*Huang and Stern, 2005*). The *Drosophila* FGF-FGFR interaction is specific. Different ligands have different functions. The activation of FGFRs by specific ligands can affect specific biological processes (*Kadam et al., 2009*). The FGFR in the membrane of Sf9 cells can bind to Vip3Aa (*Jiang et al., 2018*). One FGF and one FGFR are in the *H. armigera* genome, which have yet to be studied functionally. The study found that FGFR prevents insect pupation by transmitting JH signal as a JH cell membrane receptor. Exploring the molecular mechanism and output by

which multiple ligands transmit signals through the same receptor is exciting and challenging in future work.

CAD96CA and FGFR1 have similar functions in JH signaling, including transmitting JH signal for *Kr-h1* expression, larval status maintaining, rapid intracellular calcium increase, phosphorylation of transcription factors MET1 and TAI, and high affinity to JH III. CAD96CA and FGFR1 are essential in the JH signal pathway, and the loss-of-function of each is sufficient to trigger strong effects on pupation, suggesting they can transmit JH signal individually. The difference is that CAD96CA expression has no tissue specificity, and the *Fgfr1* gene is highly expressed in the midgut. A possibility is that CAD96CA and FGFR1 play roles by forming homodimer or heterodimer with each other or with other RTKs in tissues, which needs to be addressed in future studies. CAD96CA and FGFR1 transmit JH III signals in three different insect cell lines, suggesting their conserved roles in other insects.

Homozygous *Cad96ca* null *Drosophila* die at late pupal stages (*Wang et al., 2009*), suggesting that CAD96CA is critical to insect pupation. Here, we further revealed that *Cad96ca* mosaic mutation by CRISPR/Cas9 caused precocious pupation in *H. armigera*, suggesting that CAD96CA plays roles to prevent pupation. Similarly, null mutant of *Fgfr1* or *Fgfr2* in mouse is embryonic lethal (*Arman et al., 1998*; *Deng et al., 1994*; *Yamaguchi et al., 1994*). Htl (*Fgfr*) homozygous mutant in *D. melanogaster* die during late embryogenesis, too (*Beati et al., 2020*; *Beiman et al., 1996*; *Gisselbrecht et al., 1996*), suggesting FGFR1 is important to embryogenesis. However, in *H. armigera*, *Fgfr1* mosaic mutation mainly caused precocious pupation, suggesting FGFR1 is necessary to prevent pupation. The double mutation of *Cad96ca* and *Fgfr1* caused earlier pupation and death compared to the single mutation of *Cad96ca* or *Fgfr1*. These data suggested that both CAD96CA and FGFR1 can transmit JH signal to prevent pupation independently and cooperatively. MET1 is the intracellular receptor of JH. Knockout of *Met1* in the *B. mori* leads to precocious metamorphosis and death in the penultimate instar (*Daimon et al., 2015*). In the *Spodoptera exigua*, knockout of *Met1* also results in precocious metamorphosis and death in the penultimate and final instars (*Zhao et al., 2023*). In *D. melanogaster*, both *Met* and *Gce* null mutants die during the larval-pupal transition (*Abdou et al., 2011*). In the *H. armigera*, most of the *Met1* mutants died during the transformation from the final instar larva to the pupa, too. These data suggest that JH via MET1 prevents pupation. In the triple mutants of *Met1, Cad96ca,* and *Fgfr1*, most larvae died at the 5th and 6th instar larval stages, which is much more serious than the *Met1* mutation, or *Cad96ca* and *Fgfr1* double mutation, because of the mutation both intracellular receptor MET1 and two membrane receptors CAD96CA and FGFR1 of JH. These data suggest that JH exerts a complete regulatory role through cell membrane receptors and intracellular receptor, because the cell membrane receptors regulate the phosphorylation of the intracellular receptor MET1 and the interacting transcription factor TAI. The results from different insects suggest that JH via MET1, CAD96CA, and FGFR1 play roles in preventing metamorphosis. In *B. mori*, after the knockout of JH acid methyltransferase (*Jhamt*) using TALENs, the larvae died during L1 or L2 larval stages (*Daimon et al., 2015*). The knockdown of *Jhamt* in *Drosophila* by RNAi does not exhibit a visible effect on development (*Niwa et al., 2008*). Knockdown of *Jhamt* by RNAi in the *Tribolium* larval stage results in precocious larval-pupal metamorphosis (*Minakuchi et al., 2008a*). Homozygous *Jhamt* -/- larvae in mosquitoes hatch normally and live to the final-instar larvae (L4), and die prior to pupation (*Nouzova et al., 2021*). The results from different insects suggest that the insect development in the early larval instars is less dependent on JH. The phenotypes from *H. armigera* CAD96CA and FGFR1 editing are consistent with the general knowledge that the primary role of JH is to maintain larval status and antagonize 20E-induced metamorphosis (*Riddiford, 2008*). The insulin/insulin-like growth factor signaling (IIS) plays a major role in promoting cell proliferation and determining the larval growth rate (*Lin and Smagghe, 2019*), and 20E promotes metamorphosis (*Nijhout et al., 2014*). These hormones cross talk to regulate insect growth and metamorphic development.

The phenotypes of gene mutation in *H. armigera* are somehow different from those obtained by homozygous mutation in other animals, due to the mosaic mutation by CRISPR/Cas9. In addition, RNAi of *Cad96ca* and *Fgfr1* was observed precocious pupation as was the case in CRISPR/Cas9, suggesting the RNAi can be used for the study of gene function in insects, especially when the gene editing is embryonic lethal.

The knockdown of *Cad96ca*, *Nrk*, *Fgfr1*, and *Wsck* showed phenotypes resistant to JH III induction and the decrease of *Kr-h1* and increase of *Br-z7* expression, but knockdown of *Vegfr* and *Drl* only decreased *Kr-h1*, without increase of *Br-z7*. *Br-z7* is involved in 20E-induced metamorphosis in

*H. armigera* (*Cai et al., 2014*), whereas, *Kr-h1* is a JH early response gene that mediates JH action (*Minakuchi et al., 2009*) and represses *Br* expression (*Riddiford et al., 2010*). The high expression of *Br-z7* is possible due to the down-regulation of *Kr-h1* in *Cad96ca*, *Nrk*, *Fgfr1* and *Wsck* knockdown larvae. The different expression profiles of *Br-z7* in *Vegfr* and *Drl* knockdown larvae suggest other roles of *Vegfr* and *Drl* in JH signaling, which need further study.

This study found six RTKs that respond to JH induction by participating in JH-induced gene expression and intracellular calcium increase; however, they exert different functions in JH signaling, and finally CAD96CA and FGFR1 are determined as JH cell membrane receptors by their roles in JH-induced phosphorylation of MET1 and TAI and binding to JH III. We screened the RTKs transmitting JH signal primarily by examining some of JH-induced gene expression. By examining other genes or by other strategies to screen the RTKs might find new RTKs functioning as JH cell membrane receptors; however, the key evaluation indicators, such as the binding affinity of the RTKs to JH and the function in transmitting JH signal to maintain larval status are essential.

In addition, GPCRs also play a role in JH signaling. JH triggers GPCR, RTK, PLC, IP3R, and PKC to phosphorylate $Na^+/K^+$-ATPase-subunit, consequently activating $Na^+/K^+$-ATPase for the induction of patency in *L. migratoria* vitellogenin follicular epithelium (*Jing et al., 2018*); JH activates a signaling cascade including GPCR, PLC, extracellular $Ca^{2+}$, and PKC, which induces vitellogenin receptor (VgR) phosphorylation and promotes vitellogenin (Vg) endocytosis in *Locusta migratoria* (*Jing et al., 2021*). JH activates a signaling cascade including GPCR, Cdc42, Par6, and aPKC, leading to an enlarged opening of patency for Vg transport (*Zheng et al., 2022*). In *Tribolium castaneum*, the dopamine D2-like receptor-mediated JH signaling promotes the accumulation of vitellogenin and increases the level of cAMP in oocytes (*Bai and Palli, 2016*). In *H. armigera*, GPCRs are involved in JH III-induced broad isoform 7 (Br-Z7) phosphorylation (*Cai et al., 2014*). In summary, these published results indicate that RTKs and GPCRs contribute to JH signaling on the cell membrane; however, the GPCR functions as JH receptor needs to be addressed in the future study. The RNAi of RTKs does not affect JH-induced *Jhi-1* expression, which implies that other receptors exist, presenting a target for future study of the new JH III receptor.

## The affinity of CAD96CA and FGFR1 to JH III

RTKs are high–affinity cell surface receptors for many cytokines, polypeptide growth factors, and peptide hormones (*Trenker and Jura, 2020*). Up to now, there is no report that RTK binds lipid hormone. We determined that CAD96CA and FGFR1 have a high affinity to JH III by MST and ITC methods after they were isolated from the cell membrane.

The [$^3$H]JH III detection method is used to determine *Drosophila* MET in vitro translation product binding JH III (Kd = 5.3 nM; *Miura et al., 2005*), and *Tribolium* MET binding JH III (Kd = 2.94 nM; *Charles et al., 2011*). However, the commercial production of [$^3$H]JH III has ceased, whereas the microscale thermophoresis (MST) method is a widely used method to detect protein binding of small molecules (*Welsch et al., 2017*). Therefore, MST was used in our study as the alternative method to measure the binding strengths of RTKs with JH III. Using the MST method, we determined that the saturable specific binding of *Helicoverpa* MET1 to JH III is Kd of 6.38 nM, which is comparable to that report for *Drosophila* MET and *Tribolium* MET determined using [$^3$H]JH III, confirming MST method can be used to detect protein binding JH III. The CAD96CA exhibited saturable specific binding to JH III with a Kd of 11.96 nM, and FGFR1 showed a Kd of 23.61 nM, which is higher than that of MET1 for JH III, suggesting lower binding affinity of RTKs than the intracellular receptor MET1 for JH III. A similar phenomenon is reported in another study, the binding affinities of steroid membrane receptors are orders of magnitude lower than those of nuclear receptors (*Falkenstein et al., 2000*). NRK did not bind JH III. One possible explanation is that NRK has a low affinity to JH III and thus transmits JH signal without binding JH, or NRK requires association with other proteins to play roles. Our study provides new evidence for the binding of lipid hormones by RTK and a new method to study the binding of ligands to receptors.

We also verified the affinity of CAD96CA and FGFR1 with JH III through the ITC method, determining their respective Kd values as 79.6 and 88.5 nanomolar. ITC is a versatile analytical method for the character of molecular interactions (*Johnson, 2021*). ITC is applied in the membrane protein family, containing GPCRs, ion channels, and transporters (*Draczkowski et al., 2014*). The ITC method requires relatively high ligand and receptor concentrations for better saturation curves (*Rajarathnam*

*and Rösgen, 2014*). However, when we prepared a protein solution of 1000 nM, protein aggregation occurred, thus we used a protein solution with a concentration of 700 nM. The Kd value detected by ITC is slightly higher than the result of the MST method; the results are sufficient to confirm the high affinity of CAD96CA and FGFR1 binding to JH III.

Although JH I and JH II are natural hormones for lepidopteran larvae (*Furuta et al., 2013*; *Schooley et al., 1984*), *H. armigera* (*Liu et al., 2013*) and *B. mori* (*Deng et al., 2011*; *Kayukawa et al., 2012*) also respond to JH III. In *B. mori* Bm-aff3 cells, the effective concentrations (EC50) of JHs (JH I, JH II, JH III, JHA, or methyl farnesoate) to induce *Kr-h1* transcription are $1.6×10^{-10}$, $1.2×10^{-10}$, $2.6×10^{-10}$, $6.0×10^{-8}$, and $1.1×10^{-7}$ M, respectively (*Kayukawa et al., 2012*). In cultures of wing imaginal discs from *B. mori*, 1–2 µM JH III promotes cuticle protein 4 gene expression (*Deng et al., 2011*). The effective concentration of JH III to induce rapid calcium increase in HaEpi cells is ≥1 µM (*Wang et al., 2016*) and 500 ng of 6th instar larva (*Cai et al., 2014*). JH III is a commercially available reagent; therefore, we used JH III to carry out the experiments in this study, and the results hypothesize the possibility of CAD96CA and FGFR1 binding other JHs in addition to JH III, which should be addressed in future study.

## Relationship of cell membrane receptor and intracellular receptor

MET is determined as JH's intracellular receptor by its characters binding to JH and regulating *Kr-h1* expression (*Charles et al., 2011*; *Jindra et al., 2021*). In our study, cell membrane receptors CAD96CA and FGFR1 are also able to bind JH III and transmit JH III signal to regulate a set of JH III-induced gene expression including *Kr-h1*. Obviously, both intracellular receptor MET and cell membrane receptor CAD96CA and FGFR1 are involved in JH III signaling as receptors. JH III transmits signal by cell membrane receptor and intracellular receptor at different signaling stages, with cell membrane receptor CAD96CA and FGFR1 inducing rapid $Ca^{2+}$ signaling, which regulates the phosphorylation of MET and TAI to enhance the function of MET for gene transcription, and the intracellular receptor MET regulates gene transcription by diffusion into cells based on its lipid characteristics.

The study in human cell line HEK293 shows that overexpression of *B. mori* JH intracellular receptor MET2 and its cofactor SRC together in HEK293 cells may induce JH-dependent luciferase reporter expression through a plasmid that contains the JH specific kJHRE (JH response element containing the E-box core sequence for JH binding; *Kayukawa et al., 2012*), suggesting JH can diffuse into cells to initiate a gene expression when the insect MET2 and SRC and kJHRE exist. Our study showed that overexpressing CAD96CA or FGFR1 in HEK-293T cells elicits $Ca^{2+}$ elevation under JH III induction, suggesting CAD96CA or FGFR1 transmit a rapid signal of JH III in HEK-293T cells, which might trigger further cellular responses of HEK-293T to JH III. These data suggest that both cell membrane receptors CAD96CA and FGFR1 and intracellular receptor MET of JH can respond to JH. These proteins might be used as switches to induce a gene expression or regulate cell fate in heterogeneous cells by JH induction when the side effects are determined.

## Conclusions

CAD96CA and FGFR1 were involved in JH III signaling, including maintaining larval status, JH III-induced rapid intracellular calcium increase, gene expression, and phosphorylation of MET and TAI. CAD96CA and FGFR1 had high affinity to JH III and were possible cell membrane receptors of JH III and other JHs. CAD96CA and FGFR1 had a general role in transmitting the JH III signal for gene expression in various insect cells, suggesting their conserved roles in other insects. JH III transmits the signal by either cell membrane receptor or intracellular receptor at different stages in the signaling, with JH III transmitting the signal by cell membrane receptor CAD96CA and FGFR1 to induce rapid $Ca^{2+}$ signaling, which regulates the phosphorylation of MET and TAI to enhance the function of MET for gene transcription, and intracellular receptor MET regulates gene transcription by diffusion into cells based on its lipid characteristics. CAD96CA and FGFR1 can transmit JH signal to prevent pupation independently and cooperatively (*Figure 7*). This study presents a platform to identify the agonist or inhibitor of JH cell membrane receptors to develop an environmental-friendly insect growth regulator.

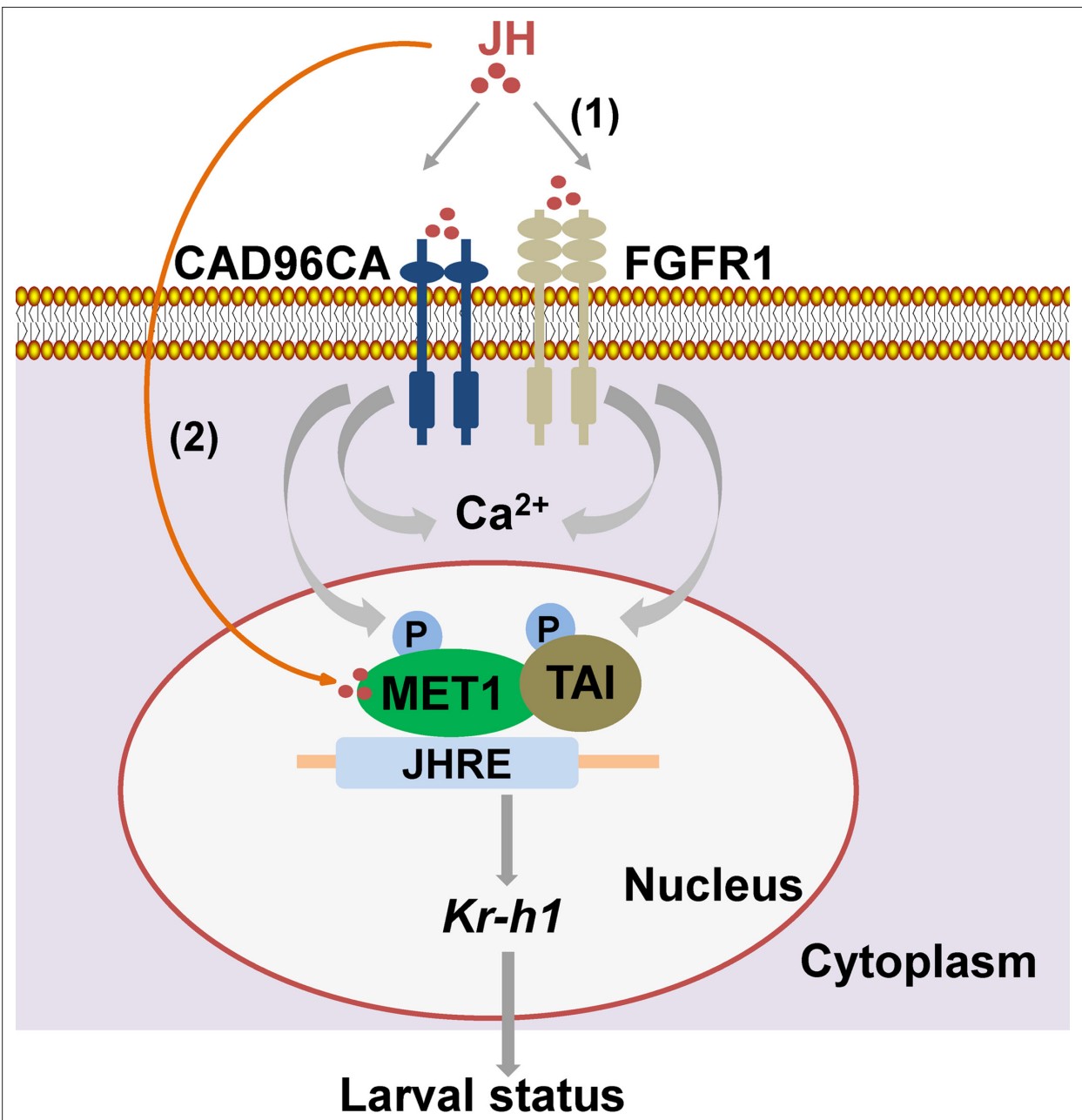

**Figure 7.** A diagram illustrating CAD96CA and FGFR1 transmitting juvenile hormone signal for gene expression. (1) JH binds to cell membrane receptors CAD96CA and FGFR1 to trigger an increase in intracellular calcium and the phosphorylation of MET1 and TAI to enhance their function in gene expression to maintain larval status. (2) On the other hand, JH enters cells freely via diffusion to bind its intracellular receptor MET. MET interacts with TAI and then binds to the JH response element (JHRE, containing the E-box core sequence, in the *Kr-h1* promoter region) to promote gene expression to maintain larval status. Therefore, JH III transmits signal by either cell membrane receptor or intracellular receptor at different stages in the signaling.

## Materials and methods

### Experimental insects

Cotton bollworms (*H. armigera*) were raised on an artificial diet comprising wheat germ and soybean powder with various vitamins and inorganic salts. The insects were kept in an insectarium at 26 ± 1 °C with 60 to 70% relative humidity and under a 14 hr light:10 hr dark cycle.

## Cell culture

Our laboratory established the *H. armigera* epidermal cell line (HaEpi) (*Shao et al., 2008*). The cells were cultured as a loosely attached monolayer and maintained at 27 °C in cell culture flasks. The cell culture flasks had an area of 25 cm² with 4 mL of Grace's medium supplemented with 10% fetal bovine serum (Biological Industries, Cromwell, CT, USA). The Sf9 cell line (Thermo Fisher Scientific, Waltham, Massachusetts, USA) was cultured in ESF921 medium at 27 °C. The S2 cell line was cultured in Schneider's *Drosophila* medium (Gibco, California, USA) with 10% FBS (Sigma, San Francisco, CA, USA) at 27 °C. The cells were subcultured when cells covered 80% of the culture flasks. The HEK-293T cell line was cultured in Dulbecco's Modified Eagle Medium (DMEM, Gibco, California, USA) with 10% FBS (Sigma, St. Louis, Missouri, USA) at 37 °C with 5% carbon dioxide.

## Bioinformatic analyses

Identification of RTKs by looking for the name of RTK in the genome of *H. armigera* using bioinformatics. Then, blast analysis was used to search for more RTKs. These RTKs were compared with previously reported RTK species in *B. mori*, *D. melanogaster*, and *H. sapiens* to confirm the amount of RTK in *H. armigera*. The phylogenetic trees were constructed from amino acid sequences using the Neighbor Joining (NJ) method in MEGA 5.0. The structure domains of the proteins were predicted using SMART (http://smart.embl-heidelberg.de/). Although the SMART tool did not predict that the TORSO has a transmembrane structure, the TORSO of *H. armigera* is 79% identity to that of TORSO of RTK members in *B. mori*. We believe that the TORSO of *H. armigera* belongs to the RTK family, but SMART failed to predict its structure successfully. Although the SMART tool did not predict the complete structure of STE20-like, it was clustered with the RTK of *D. melanogaster* Tie-like in evolutionary tree clustering analysis. In addition, in sequence alignment, the named flocculation protein FLO11-like in *Hyposmocoma kahamanoa* was 85% identity to it, and FLO11-like protein showed transmembrane structure in domain prediction, so the STE20-like of *H. armigera* was classified as a member of the RTK family.

## Double-stranded RNA synthesis

RNA interference (RNAi) has been used widely in moths of 10 families (*Xu et al., 2016*). Long double-stranded RNA (dsRNA) can be processed into smaller fragments, with a length of 21–23 nucleotides (*Zamore et al., 2000*), to restrain transcription of the target gene (*Fire et al., 1998*). dsRNA transcription was performed as follows: 2 μg of DNA template, 20 μL of 5×transcription buffer, 3 μL of T7 RNA polymerase (20 U/μL), 2.4 μL of A/U/C/GTP (10 mM) each, 3 μL of RNase inhibitor (40 U/μL, Thermo Fisher Scientific, Waltham, USA), and RNase-free water were mixed to a volume of 50 μL. After incubation at 37 °C for 4–6 hr, 10 μL RNase-free DNase I (1 U/μL, Thermo Fisher Scientific), 10 μL of DNase I Buffer, and 30 μL RNase-free water were added to the solution, which was incubated at 37 °C for 1 hr. The solution was extracted with phenol/chloroform and precipitated with ethanol; the precipitate was resuspended with 50 μL RNase-free water. The purity and integrity of the dsRNA was determined using agarose gel electrophoresis. A MicroSpectrophotometer (GeneQuant; Amersham Biosciences, Little Chalfont, UK) was used to quantify the dsRNAs.

## RNA interference in HaEpi cells

When the HaEpi cell density reached 70 to 80% in six-well culture plates, the cells were transfected with dsRNA (1 μg/mL) and Quick Shuttle Enhanced transfection reagent (8 μL; Biodragon Immuno-technologies, Beijing, China) diluted in sterilized saline medium (200 μL), and incubated with Grace's medium. The cells were cultivated for 48 hr at 27 °C. After that, the medium was replaced with a fresh Grace's medium with JH III at a final concentration of 1 μM for 12 hr. An equivalent volume of DMSO was a control. The total mRNA was then extracted for qRT-PCR.

## RNA interference in larvae

The DNA fragments of *Rtks* were amplified as a template for dsRNA synthesis using the primers RTK-RNAiF and RTK-RNAiR (*Supplementary file 2*). The dsRNAs (*dsRtk*, *dsGFP*) were injected using a micro-syringe into the larval hemocoel of the fifth instar 20 hr at 500 ng/larva, using three injections at 36 hr intervals. At 12 hr after the last injection, 500 ng of JH III (Santa Cruz Biotechnology, Santa Cruz, CA, USA) was injected into each larva. Dimethyl sulfoxide (DMSO) was used as a control. The

phenotypes and developmental rates of the larvae were recorded. The mRNA was isolated from the larvae at 12 hr after JH III injection.

## Protein overexpression

The nucleotide sequence of the genes involved in this study was cloned into the pIEx-4-His, pIEx-4-GFP-His, pIEx-4-CopGFP-His, pcDNA3.1-GFP-His or pcDNA3.1-His vector. The cells were cultured to 80% confluence at 27 °C or 37 °C in the medium. For transfection, approximately 5 µg of plasmids, 200 µL of sterilized saline water medium or Opti-MEM medium, and 8 µL of transfection reagent (Biodragon, Beijing, China) were mixed with the cells in the medium for 24–48 hr.

## Quantitative real–time reverse transcription PCR (qRT–PCR)

Total RNA was extracted from HaEpi cells and larvae using the Trizol reagent (TransGen Biotech, Beijing, China). According to the manufacturer's instructions, first-strand cDNA was synthesized using a 5×All-In-One RT Master Mix (Abm, Vancouver, Canada). qRT–PCR was then performed using the CFX96 real–time system (Bio-Rad, Hercules, CA, USA). The relative expression levels of the genes were quantified using *Actb* (β-actin) expression as the internal control. The primers are listed in *Supplementary file 2*. The experiments were conducted in triplicate with independent experimental samples. The relative expression data from qRT–PCR were calculated using the formula: $R=2^{-\Delta\Delta CT}$ ($\Delta\Delta Ct = \Delta Ct_{sample}-\Delta Ct_{control}$, $\Delta Ct = Ct_{gene}-Ct_{\beta-actin}$; *Livak and Schmittgen, 2001*).

## Detection of the cellular levels of calcium ions

The cells were cultured to a density of 70–80%. The cells were incubated with Dulbecco's phosphate-buffered saline (DPBS; 137 mM NaCl, 2.7 mM KCl, 1.5 mM $KH_2PO_4$, and 8 mM $Na_2HPO_4$) including 3 µM acetoxymethyl (AM) ester calcium crimson dye (Invitrogen, Carlsbad, CA, USA) for 30 min at 27 °C. The cells were washed with fresh DPBS three times. The cells were then exposed to 1 µM JH III to detect the intracellular calcium release. After that, cells were treated with Calcium chloride (final concentration 1 mM) and JH III (final concentration 1 µM), and put into a microscope dish. Fluorescence was detected at 555 nm, and the cells were photographed automatically once every 6 s for 420 s using a Carl Zeiss LSM 700 laser scanning confocal microscope (Thornwood, NY, USA). The fluorescence intensity of each image was analyzed using Image Pro-Plus software (Media Cybernetics, Rockville, MD, USA).

## Western blotting

Epidermis, midgut, and fat body tissues were homogenized in 500 µL Tris-HCl buffer (40 mM, pH 7.5) on ice with 5 µL phenylmethylsulfonyl fluoride (PMSF, 17.4 mg/mL in isopropyl alcohol), respectively. The homogenate was centrifuged for 15 min at 4 °C at 12,000× *g*, then supernatant was collected. The protein concentration in the supernatant was measured using the Bradford protein assay. Proteins (20 µg per sample) sample was subjected to 7.5% or 12.5% SDS-PAGE and transferred onto a nitrocellulose membrane. The membrane was incubated in blocking buffer (Tris-buffered saline, 150 mM NaCl, 10 mM Tris-HCl, pH 7.5, with 3–5% fat-free powdered milk) for 1 h at room temperature. The primary antibody was diluted in blocking buffer, then incubated with the membrane at 4 °C overnight. The membrane was washed three times wash with TBST (0.02% tween in TBS) for 10 min each. Subsequently, the membrane was incubated with secondary antibodies, 1:10,000 diluted, alkaline phosphatase-conjugated (AP) or horseradish peroxidase-conjugated (HRP) AffiniPure Goat Anti-Rabbit/-Mouse IgG (ZSGB-BIO, Beijing, China). The membrane was washed twice with TBST and once with TBS. The immunoreactive protein bands marked by AP were observed after incubating in 10 mL of TBS solution combined with 45 µL of P-nitro-blue tetrazolium chloride (NBT, 75 µg/µL) and 30 µL of 5-bromo-4-chloro-3 indolyl phosphate (BCIP, 50 µg/µL) in the dark for 10–30 min. The reactions were stopped by washing the membrane with deionized water and images by the scanner. The proteins marked by HRP were detected using a High-Sig ECL Western Blotting Substrate and exposed to a Chemiluminescence imaging system (Tanon, Shanghai, China), according to the manufacturer's instructions. The immunoreactive protein band density was calculated using ImageJ software (National Institutes of Health, Bethesda, MD, USA). The data were analyzed using GraphPad Prism 5 software (GraphPad Software, San Diego, CA, USA).

## Lambda protein phosphatase (λPPase) treatment

The protein suspension (40 µL, 0.1 mg/mL) was incubated with $\lambda$ PPase (0.5 µL), buffer (5 µL), and MnCl$_2$ (5 µL) at 30 °C for 30 min, according to the manufacturer's specifications (New England Biolabs, Beijing LTD, Beijing, China). Total proteins were subjected to SDS-PAGE and then electrophoretically transferred onto a nitrocellulose membrane for western blotting.

## Phos-tag SDS-PAGE

Phos-tag Acrylamide (20 µM; Fujiflm Wako Pure Chemical Corporation, Osaka, Japan) and MnCl$_2$ (80 µM) were mixed into a normal SDS-PAGE gel. The phosphates of the phosphorylated protein can bind to Mn$^{2+}$, which reduces the mobility of the phosphorylated protein in the gel. The protein sample was treated with 20% trichloroacetic acid (TCA) to remove the chelating agent. The gel was shaken and incubated three times in 10 mmol/L EDTA transfer buffer solution for Phos-tag SDS-PAGE for 10 min each time. Mn$^{2+}$ was removed, and then the proteins were electrophoretically transferred to a nitrocellulose membrane and analyzed using western blotting.

## Immunocytochemistry

The cells were grown on coverslips, treated with hormones, washed three times with DPBS, and fixed using 4% paraformaldehyde in PBS for 10 min in the dark. The fixed cells were incubated with 0.2% Triton-X 100 diluted in PBS for 10 min. The cells were washed with DPBS five times for 3 min each, and the plasma membrane was stained with Alexa Fluor 594-conjugated wheat germ agglutinin (WGA) (1:2,000 in PBS) (Invitrogen, Carlsbad, CA, USA) for 8 min. The cells were washed with DPBS five times for 3 min each, and stained with 4', 6-diamidino-2-phenylindole (DAPI, 1 µg/mL in PBS; Sigma, San Francisco, CA, USA) in the dark at room temperature for 8 min. The fluorescence signal was detected using an Olympus BX51 fluorescence microscope (Olympus, Tokyo, Japan). Scale bar = 20 µm.

## Mutations of CAD96CA and FGFR1

The structures of CAD96CA and FGFR1 were predicted online with SMART. According to the location of the predicted domain, the target fragment was amplified with mutated primers (*Supplementary file 2*) and cloned into the pIEx-4-CopGFP-His, pcDNA3.1-GFP-His or pcDNA3.1-His vector. The CAD96CA mutants were constructed to CAD96CA-M1-CopGFP-His (AA: 51–615), CAD96CA-M2-CopGFP-His (AA: 101–615), CAD96CA-M3-CopGFP-His (AA: 151–615), and CAD96CA-M4-CopGFP-His (AA: 201–615). FGFR1 mutants were constructed to FGFR1-M1-CopGFP-His (AA: 101–852), FGFR1-M2-CopGFP-His (AA: 201–852), FGFR1-M3-CopGFP-His (AA: 301–852), and FGFR1-M4-CopGFP-His (AA: 401–852).

## Detection of RTK binding JH III by microscale thermophoresis

RTKs and MET1 were recombined in plasmid pIEx-4-CopGFP-His, which was overexpressed in Sf9 cells. After 48 hr, total plasma membrane RTKs were extracted using a cell transmembrane protein extraction kit (BestBio, Shanghai, China). MET1-CopGFP-His and CopGFP-His were extracted using radioimmunoprecipitation assay (RIPA) lysis buffer (20 mM Tris-HCl, pH 7.5; 150 mM NaCl; and 1% Triton X-100) without ethylenediaminetetraacetic acid (EDTA; Beyotime, Shanghai, China). A 100 µL of slurry of chelating Sepharose with Ni$^{2+}$ was washed three times with binding buffer (500 mM NaCl; 20 mM Tris-HCl, pH 7.9; and 5 mM imidazole) for 5 min. The overexpressed proteins were bound to the washed Ni$^{2+}$-chelating Sepharose (GE Healthcare, Pittsburgh, PA, USA). The suspension was mixed on a three-dimensional rotating mixer for 40 min at 4 °C. Then, the resin was washed three times for 5 min each time with wash buffer (0.5 M NaCl; 20 mM Tris-HCl, pH 7.9; and 20 mM imidazole). After centrifugation at 500×*g* for 3 min at 4 °C, the RTKs were washed three times with wash buffer for 5 min each time. The RTKs were eluted using 100 µL of elution buffer (0.5 M NaCl; 20 mM Tris-HCl, pH 7.9; 100 mM imidazole; and 0.5% Triton X-100) and then diafiltration was carried out three times with PBST (PBS, 0.05% Tween, and 0.5% Triton X-100) buffer using Amicon Ultra 0.5 (Merck Millipore, Temecula, CA, USA) to reduce the concentration of imidazole in preparation for the subsequent experiment. The concentration of the isolated RTK was detected using a BCA protein assay kit (Beyotime, Shanghai, China). JH III bound by 50 nM RTK was detected using the microscale thermophoresis (MST) method (*Huang and Zhang, 2021*; *Welsch et al., 2017*). Firstly, the fluorescence intensity and the homogeneity of the protein solution were detected. We confirmed that the fluorescence intensity

of the protein samples was within the range of the instrument, and there was no aggregation of the protein samples. Then, we carried out experiments. 16 microtubes were prepared, and the ligand was diluted for use at the initial concentration of 1 µM JH III. Specifically, 5 µL of the ligand buffer was added to prepared microtubes No. 2–16. After, 10 µL of the ligand was added to tube No. 1, 5 µL of the ligand solution in tube No. 1 was pipetted out of tube No. 1, added to tube No. 2, and mixed well. Then 5 µL of solution was pipetted from tube No. 2 and added to tube No. 3. Finally, 5 µL of mixed liquid was removed from tube No. 16 and discarded. (The original concentration of JH III was dissolved in DMSO, and therefore, DMSO needed to be added to the ligand dilution buffer to ensure an equal amount of DMSO in each tube). Then, 5 µL of the fluorescence molecule (target protein) was added to each tube and mixed well. With each tube holding a 10 µL volume in total, the tubes were incubated at 4 °C for 30–60 min. Finally, samples were removed with a capillary tube and tested with an MST Monolith NT.115 (NanoTempers, Munich, Germany).

## Detection of RTK binding JH III by isothermal titration calorimetry

The protein purification method was described in the MST experiment. The isothermal titration calorimetry (ITC) assay was performed using MicroCal PEAQ-ITC (Malvern Panalytical, Malvern, U.K.). JH III was dissolved in ethanol, JH III stock solution to a final concentration of 10 µM with PBST buffer. The protein solution with same concentration ethanol, make sure the buffer identity. According to the manufacturer's instructions, JH III (10 µM) was loaded in a syringe, and the protein solution (700 nM) was injected into the ITC cell. Injection of 3 µl of JH III solution over a period of 150 s at a stirring speed of 750 rpm was performed. For the control test, JH III solution was pumped into syringe, and the buffer was injected into the ITC cell. For the data, the experimental data were subtracted with that from the control test by analysis software.

## Methyl farnesoate, farnesol, methoprene binding assays, and competition assays

Methyl farnesoate (Echelon Biosciences, Utah, USA), farnesol (Sigma, San Francisco, CA, USA), and methoprene (Sigma, San Francisco, CA, USA) were dissolved in DMSO, respectively, diluted to the corresponding concentration, and the experimental method as described by the MST method for detection of binding. The competitive binding by MST requires fluorescent labeling of ligands (JH III). Currently, there is no suitable method to label JH III, and we only have fluorescently labeled receptors (target protein). The binding curve of adding both JH III and methoprene, but the maximum concentration of JH used in the experiment was 50 nM, while the concentration of methoprene was increasing. The Kd value is generated automatically by the software of the instrument.

## CRISPR/Cas9 system mediated knockout of *Cad96ca*, *Fgfr1* or *Met1*

The gRNAs were designed using the CRISPRscan tool (https://www.crisprscan.org/?page=sequence) (*Zhang et al., 2021*) and each consisted of an ~20-nucleotide (nt) region in complementary reverse to one strand of the target DNA (protospacer) with an NGG motif at the 3' end (PAM) of the target site and a GGN at position (5' end) of the T7 promoter. The sgRNA primer and universal primer were used as corresponding templates to obtain amplification products. Product transcription was carried out with a T7 Transcription Kit (Thermo Fisher Scientific, Waltham, USA) following the manufacturer's instructions.

Freshly laid eggs on gauze (within 2 hr) were collected from gauze using 0.1% (v/v) 84 solution and rinsed with distilled water. The eggs were affixed onto microscope slides using double-sided adhesive tape (*Zuo et al., 2017*; *Zuo et al., 2018*). A mixture of 100 ng/µL Cas9 protein (GenScript, New Jersey, USA) and 300 ng/µL gRNA for the injection into the eggs (per egg 2 nL was injected) within 4 hr of oviposition using a Pico-litre Microinjector (Warner Instruments, Holliston, USA) (*Hou et al., 2021*). The injected eggs were incubated at 26 ± 1 °C with 60 to 70% relative humidity for 3–4 days until they hatched. To detect the mutagenesis of *H. armigera* induced by CRISPR/Cas9, we used PCR to amplify the targeted genomic region obtained from fresh epidermis samples of larvae moulted from G0 individuals and used primers at approximately 50–200 base pairs upstream and downstream from the expected double strand break site by HiFi DNA Polymerase (Transgen, Beijing, China). For the wild-type group, 10 individuals were randomly selected for PCR products sequencing analysis. For the experimental group, individuals with the phenotypes of early pupation

and dead were all subjected to sequencing analysis of PCR products. For the experimental group, individuals without significant phenotypes were randomly divided into three groups, and the mixed samples were used for sequencing analysis. The sequencing results showed overlapping peaks at the target sites, and these individuals were identified as mutants. The corresponding PCR products of 10 randomly selected mutant individuals were cloned into a pMD19-T vector (TaKaRa, Osaka, Japan) for sequencing, respectively. Six colonies from one individual were randomly picked for sequencing to identify the rates of mutation in an individual. The mutated sites were identified by comparison with the wild-type sequence. The ratio of the mutants in the PCR products was the editing efficiency. The editing efficiency of the *Cad96ca* mutant was 67%, that of the *Fgfr1* mutant was 61%, that of the *Met1* mutant was 72%, that of the *Cad96ca* and *Fgfr1* mutant was 69%, and that of the *Met1*, *Cad96ca* and *Fgfr1* mutant was 70%. To detect off-target activity of the CRISPR/Cas9 system-created *Cad96ca* and *Fgfr1* mutants, we searched the *H. armigera* genome for homologues of the target sequences of *Cad96ca* and *Fgfr1* and found that the genes possibly included similar target sequences. PCR amplification and sequencing were performed with these genes.

## Generation of *Cad96ca*- or *Fgfr1*-mutant HaEpi cells using the CRISPR/Cas9 system

The target sites were selected according to the CRISPRscan tool (*Supplementary file 2*). Then, two complementary oligonucleotides were synthesized according to the target sequences, and the annealed fragments were cloned into a pUCm-T-U6-gRNA plasmid after forming double chains. Primers gRNAwf-F and gRNAwf-R were used for PCR amplification with the pUCm-T-U6-gRNA plasmid carrying with target gRNA sequence as a template. The obtained fragment was cloned into a pIEx-Cas9-GFP-P2A-Puro plasmid, and pIEx-4-BmU6-gRNA-Cas9-GFP-P2A-Puro was successfully constructed. The pIEx-4-BmU6-Cad96ca-gRNA-Cas9-GFP-P2A-Puro or pIEx-4-BmU6-*Fgfr1*-gRNA-Cas9-GFP-P2A-Puro recombinant vectors were transfected into HaEpi cells with transfection reagent (Roche, Basel, Switzerland). After 48 hr of vector transfection (cells can be observed to express green fluorescent protein), fresh medium containing puromycin (Solarbio, Beijing, China; 15 μg/mL) was added to the cells, the medium containing puromycin was replaced every 2 days until the green fluorescence was gone (about 5 days), and the medium was replaced. The puromycin-screened cells were used for subsequent experiments. Messy peak figures reporting the results of DNA sequencing showed mutations induced by CRISPR/Cas9 in the HaEpi cells.

## Detection of the cellular levels of calcium ions as indicated by protein calcium-sensing GCaMPs

GCaMPs are the most widely used protein calcium sensors (*Dana et al., 2019*). The CMV promoter of pCMV-GCaMP5G was replaced with an IE promoter and transformed into pIE-GCaMP5G, which can be expressed in HaEpi cells. pIE-GCaMP5G was transfected into normal HaEpi cells, *Cad96ca*- and *Fgfr1*-mutant HaEpi cells for 48 hr and incubated with JH III (1 μM) or JH III (1 μM) plus CaCl$_2$ (1 mM) for 60 s. First, the cells were photographed in white light and then imaged with a fluorescence microscope.

## Calcium levels were detected by Flow-8 AM fluorescence probe

Intracellular calcium levels in Sf9 cells, S2 cells, and HEK-293T cells were determined using the fluorescent probe Fluo-8 AM (MKBio, Shanghai, China). Cells were seeded overnight at 50,000 cells per 100 μL per well in a 96-well black wall/clear bottomed plate. The Fluo-8 dye was diluted to 2 μM with DPBS, while the 20% PluronicF-127 solution was added for a final concentration of 0.02%. Add 100 μl Fluo-8 dye solution to each well. Then the plate was incubated at room temperature for 30 min. The cells were washed with DPBS three times. After 60 s stabilization of the cells, JH III (1 μM) was added to the cells to detect the intracellular calcium release to show the cells can respond JH III induction. After that, cells were treated with Calcium chloride (final concentration 1 mM) and JH III (final concentration 1 μM) to detect the extracellular calcium influx. DMSO was a solvent control of JH III. The fluorescence intensities were measured using an ENSPIE plate reader (PE, New York, USA) with a filter set of Ex/Em = 490/514 nm.

### Antibodies

The following antibodies were used in the study: anti-His monoclonal antibody (1:3000, AE003, RRID:AB_2728734, ABclonal, Wuhan, China), anti-GFP monoclonal antibody (1:3000, AE012,

RRID:AB_2770402, ABclonal, Wuhan, China), anti-ACTB polyclonal antibodies (1:3000, AC026, RRID:AB_2768234, ABclonal, Wuhan, China).

## Statistical analysis

All data were from at least three biologically independent experiments. The western blotting results were quantified using ImageJ software (NIH, Bethesda, MA, USA). The fluorescence intensity of each image of calcium detection was analyzed using Image Pro-Plus software (Media Cybernetics, Rockville, MD, USA). GraphPad Prism 7 was used for data analysis and producing figures (GraphPad Software Inc, La Jolla, CA, USA). Multiple sets of data were compared by ANOVA. The different lowercase letters show significant differences. Two group datasets were analyzed using a two-tailed Student's $t$ test. Asterisks indicate significant differences between the groups (*$p<0.05$, **$p<0.01$). Error bars indicate the SD or SE of three independent experiments.

## Acknowledgements

We thank Jingyao Qu, Zhifeng Li, and Jing Zhu at the State Key Laboratory of Microbial Technology, Shandong University for their help in using MST Monolith NT.115. We thank Xiangmei, Ren at the State Key Laboratory of Microbial Technology, Shandong University for help with using ENSPIE plate reader. We thank Feng Zhang from the Core Facility and Service Platform at the School of Life Sciences, Shandong University for help with using BioTek plate reader. This study was supported by the National Natural Science Foundation of China (grant nos. 32330011 and 32270507).

## Additional information

### Funding

| Funder | Grant reference number | Author |
| --- | --- | --- |
| National Natural Science Foundation of China | 32330011 | Xiao-Fan Zhao |
| National Natural Science Foundation of China | 32270507 | Xiao-Fan Zhao |

The funders had no role in study design, data collection and interpretation, or the decision to submit the work for publication.

### Author contributions

Yan-Xue Li, Conceptualization, Data curation, Investigation, Visualization, Methodology, Writing – original draft; Xin-Le Kang, Software, Investigation; Yan-Li Li, Xiao-Pei Wang, Qiao Yan, Investigation; Jin-Xing Wang, Conceptualization, Writing – review and editing; Xiao-Fan Zhao, Conceptualization, Funding acquisition, Writing – original draft, Writing – review and editing

### Author ORCIDs

Yan-Xue Li https://orcid.org/0000-0002-4675-4954
Jin-Xing Wang https://orcid.org/0000-0003-0283-3930
Xiao-Fan Zhao https://orcid.org/0000-0003-1809-4730

Reviewer #2 (Public review): https://doi.org/10.7554/eLife.97189.4.sa1
Author response https://doi.org/10.7554/eLife.97189.4.sa2

## Additional files

### Supplementary files

Supplementary file 1. Names of RTKs identified in *H. armigera* genome.

Supplementary file 2. Oligonucleotide sequences of PCR primers.

MDAR checklist

## Data availability

All data generated or analysed during this study are included in the manuscript and source data files; source data files have been provided for all figures.

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
