## [Editor Report · eLife Assessment]

In this **important** study, Li and others identified cell membrane receptors for juvenile hormone (JH), a terpenoid hormone in insects that regulates their development and reproduction. While intracellular receptors for JH are well characterized, membrane receptors for JH have remained elusive for many years. The authors provide **convincing** evidence indicating that two receptor tyrosine kinases (RTKs), CAD96CA and FGFR1, modulate the genomic effects of JH by phosphorylating the intracellular receptors in the cotton bollworm, Helicoverpa armigera. Although differential functions of the two RTKs and potential effects of the other endogenous ligands of these RTKs on JH signaling remain unclear, this study lays a foundation for future studies.

---

## [Referee Report · Reviewer #2 (Public review)]

Summary:

Juvenile hormone (JH) is a pleiotropic terpenoid hormone in insects that mainly regulates their development and reproduction. In particular, its developmental functions are described as the "status quo" action, as its presence in the hemolymph (the insect blood) prevents metamorphosis-initiating effects of ecdysone, another important hormone in insect development, and maintains the juvenile status of insects.

While such canonical functions of JH are known to be mediated by its intracellular receptor complex composed of Met and Tai, there have been multiple reports suggesting the presence of cell membrane receptor(s) for JH, which mediate non-genomic effects of this terpenoid hormone. In particular, the presence of receptor tyrosine kinases (RTKs) that phosphorylate Met/Tai in response to JH and thus indirectly affect the canonical JH signaling pathway has been strongly suggested. Given the importance of JH in insect physiology and the fact that the JH signaling pathway is a major target of insect growth regulators, elucidating the identify and functions of putative JH membrane receptors is of great significance form both basic and applied perspectives.

In the present study, the authors identified candidate receptors for such cell membrane JH receptors, CAD96CA and FGFR1, in the cotton bollworm, Helicoverpa armigera.

Strengths:

Their in vitro analyses are conducted thoroughly using multiple methods, which overall support their claim that these receptors can bind to JH and mediate their non-genomic effects.

Their CRISPR-Cas-mediated mutagenesis in vivo shows that mutation of the two RTKs causes acceleration of pupation, which is consistent with the mutant phenotype of the intracellular JH receptor, Met1. Although this is different from the typical phenotype one would expect from JH signaling deficiency in lepidopteran insects (i.e. precocious metamorphosis), the results overall support their claim that these two RTKs modulate genomic JH effects by phosphorylating the intracellular receptors.

Weaknesses:

Although their loss-of-function analyses suggest that the two RTKs likely have redundant functions in vivo, it is unclear whether they have any different functions in mediating JH functions in different physiological contexts. It also remains unknown whether other endogenous ligands for these RTKs affect canonical, genomic JH signaling in vivo.

---

## [Author Response]

The following is the authors’ response to the previous reviews.

**Public Reviews:**

**Reviewer #1 (Public review):**
[…] Weaknesses:Unfortunately, the revised manuscript does not show significant improvement. While the identification of the receptors is highly convincing, important issues about the biological relevance remain unaddressed. First, the main point I raised about the first version of this article is that the redundancy and/or specificity of the two receptors should be clarified, even though I understand that it cannot be deeply investigated here. I believe that this point, shared by all reviewers, is highly relevant for the scope of this work. In this revised version, it is still unclear how to reconcile gain and loss-of-function experiments and the different expression profiles of the receptors. Second, the newly added explanations and pieces of discussion provided about the mild in vivo phenotypes of early pupation upon Cad96ca or Fgfr1 knock-out do not clarify the issue but instead put emphasis on methodological issues. Indeed, it is not clear whether the mild phenotypes reflect the biological role of Cad96ca and Fgfr1, or the redundancy of these two RTKs (and/or others), or some issue with the knock-out strategy (partial efficiency, mosaicism...). Finally, parts of the updated discussion and the modifications to the figures are confusing.

Thank you for asking the questions. We performed additional experiments, including editing *Met1* individually (single knockout), *Cad96ca* and *Fgfr1* together (double knockout), and *Met1*, *Cad96ca* and *Fgfr1* together (triple knockout) using CRISPR/Cas9. The results showed that single mutation of *Cad96ca* or *Fgfr1* caused precocious pupation, respectively. The double mutation of *Cad96ca* and *Fgfr1* caused earlier pupation and death compared to the single mutation of *Cad96ca* or *Fgfr1*. The triple mutation of *Met1*, *Cad96ca* and *Fgfr1* caused most serious effect on pupation time and death. These data suggested that both CAD96CA and FGFR1 can transmit JH signal to prevent pupation independently and cooperatively, and the JH exert a complete regulatory role through cell membrane receptors and intracellular receptor of JH. We increased the results in Lines 242-263 and discussion in Lines 328-375.

CAD96CA and FGFR1 have similar functions in JH signaling, including transmitting JH signal for *Kr-h1* expression, larval status maintaining, rapid intracellular calcium increase, phosphorylation of transcription factors MET1 and TAI, and high affinity to JH III. CAD96CA and FGFR1 are essential in the JH signal pathway, and the loss-of-function of each is sufficient to trigger strong effects on pupation, suggesting they can transmit JH signal individually. The difference is that CAD96CA expression has no tissue specificity, and the *Fgfr1* gene is highly expressed in the midgut. A possibility is that CAD96CA and FGFR1 play roles by forming homodimer or heterodimer with each other or with other RTKs in tissues, which needs to be addressed in future studies. CAD96CA and FGFR1 transmit JH III signals in three different insect cell lines, suggesting their conserved roles in other insects.

The mild phenotypes shown in the previous picture, Fig 4E, were counted from all the surviving individuals injected with gRNA, including mutated and non-mutated individuals. In fact, there is no phenotype of pupation on time in the mutants. According to the first round of reviewers' comments, we found that it was inappropriate to count all the surviving individuals injected with gRNA, so we replaced the picture by counting the phenotypes of all successfully mutated individuals in the second version to avoid the confusion of the phenotypes.

**Reviewer #2 (Public review):**
[…] Weaknesses:Results of their in vivo experiments, particularly those of their loss-of-function analyses using CRISPR mutants are still preliminary, and the results rather indicate that these membrane receptors do not have any physiologically significant roles in vivo. More specifically, previous studies in lepidopteran species have clearly and repeatedly shown that precocious metamorphosis is the hallmark phenotype for all JH signaling-deficient larvae. In contrast, the present study showed that Cad96ca and Fgfr1 G0 mutants only showed slight acceleration in their pupation timing, which is not a typical phenotype one would expect from JH signaling deficiency. This is inconsistent with their working model provided in Figure 6, which indicates that these cell membrane JH receptors promote the canonical JH signaling by phosphorylating Met/Tai. If the authors argue that this slight acceleration of pupation is indeed a major JH signaling-deficient phenotype in Helicoverpa, they need to provide more data to support their claim by analyzing CRISPR mutants of other genes involved in JH signaling, such as Jhamt and Met. An alternative explanation is that there is functional redundancy between CAD96CA and FGFR1 in mediating phosphorylation of Met/Tai. This possibility can be tested by analyzing double knockouts of these two receptors. Currently, the validity of their calcium imaging analysis in Figure 5 is also questionable. When performing calcium imaging in cultured cells, it is critically important to treat all the cells at the end of each experiment with a hormone or other chemical reagents that universally induce calcium increase in each particular cell line. Without such positive control, the validity of calcium imaging data remains unknown, and readers cannot properly evaluate their results.

Thank you for the comments. We took your suggestions and performed additional experiments, editing *Met1* individually (single knockout), *Cad96ca* and *Fgfr1* together (double knockout), and *Met1*, *Cad96ca* and *Fgfr1* together (triple knockout) using CRISPR/Cas9. We increased the results in Lines 242-263 and discussion in Lines 328-375.

About the calcium imaging in cultured cells (now Fig 6), our goal is to examine the roles of CAD96CA and FGFR1 in JH trigged cellular responses. The experiment was well designed and controlled and the results were validated. For examples: JH III induced intracellular Ca^2+^ release and extracellular Ca^2+^ influx in Sf9 and S2 cells, but DMSO could not. However, knockdown of *Cad96ca* and *Fgfr1* significantly decreased JH III-induced intracellular Ca^2+^ release and extracellular Ca^2+^ influx (Figure 6A, B), and *Kr-h1* expression (Figure 6—figure supplement 1A and B), suggesting that CAD96CA and FGFR1 had a general function to transmit JH signal in *S. frugiperda* and *D. melanogaster*.

Wild mammalian HEK-293T cells had no significant changes in calcium ion levels under JH III induction, because there is no CAD96CA and FGFR1 in mammal cells (Figure 6C). However, when HEK-293T cells were overexpressed insect CAD96CA or FGFR1, respectively, JH III triggered rapid cytosolic Ca^2+^ release and influx (Figure 6D).

An increase in Ca^2+^ was not detected in mutants of CAD96CA-M3 and CAD96CA-M4 under JH III induction (Figure 6E) and nor in FGFR1-M4 (Figure 6F). These results confirmed that CAD96CA and FGFR1 play roles in transmitting JH III signal.

We carefully revised the description of the results and methods to help people understand the study.

**Reviewer #3 (Public review):**
[…] Weaknesses:The authors have provided evidences that the Cad96Ca and FGF1 RTK receptors contribute to JH signaling through CRISPR/Cas9, inducing precocious metamorphosis, although not to the same extent as absence of JH. Therefore, it still remains unclear whether these RTKs are completely required for pathway activation or only necessary for high activation levels during the last larval stage. While the authors have included some additional data, the mechanism by which different RTKs function in transducing JH signaling in a tissue specific manner is still unclear. As the authors note in the discussion, it is possible that other RTKs may also play a role in facilitating the transduction of JH signaling. Lastly, the study does not yet explain how RTKs with known ligands could also bind JH and contribute to JH signaling activation. Although receptor promiscuity has been suggested as a possible mechanism, future studies could explore whether activation of RTK pathways by their known ligands induces certain levels of JH transducer phosphorylation, which, in the presence of JH, could contribute to full pathway activation without the need for direct JH-RTK binding.

Thank you for your comments. To address your questions, we carried out additional experiments. The relevant results have been incorporated into Lines 242-263, and the corresponding discussion has been added to Lines 328-375.

We agree with your suggestions that the future studies should resolve the questions such as how different RTKs function in transducing JH signaling in a tissue specific manner; whether other RTKs can transduce JH signal; how RTKs with known ligands could also bind JH and contribute to JH signaling activation; and how the RTK pathways are activated by their ligands.

**Recommendations for the authors:**

**Reviewer #1 (Recommendations for the authors):**
(1) First, some of the new paragraphs, repeatedly used in the point-by-point answer to the reviewers, are highly confusing and need proofreading (i.e. 225-230; 320-340)

Thank you for your advice. We have carefully revised the manuscript and the point-by-point answer to avoid repetition.

(2) While the double knock-down or knock-out of Cad96ca and Fgfr1 is expected to provide valuable information regarding their respective functions, the authors indicated that they wouldn't provide experiments in that direction. It is not clear to me if they have tried or not. The Crispr/Cas9 approach might be difficult to put in place to test this interaction. However, couldn't the authors try the double knock-down compared to single knock-downs using dsRNA? This method gave convincing results to test the role of the putative receptors in mediating JH-induced developmental delay in vivo (Figure 1).

Thank you for your suggestion. We added experiments, editing *Met1* individually (single knockout), *Cad96ca* and *Fgfr1* together (double knockout), and *Met1*, *Cad96ca* and *Fgfr1* together (triple knockout) using CRISPR/Cas9, the new evidence fully defined the physiological roles of these receptors in JH signaling in vivo. We increased the results in Lines 242-263 and discussion in Lines 328-375.

(3) Concerning the effect of Crispr knock-out on pupation timing, this paragraph was added: "The low death rate after Cad96ca and Fgfr1 knockout might be because of following reasons, including the editing efficiency (67% and 61% for Cad96ca mutant and Fgfr1 mutant, respectively), the chimera of the gene knockout at the G0 generation, and the redundant RTKs that play similar roles in JH signaling". A similar explanation applies to the pupation phenotype itself... I am therefore wondering whether the Crispr/Cas9 approach (at the G0 generation) is the best strategy. Since the dsRNA knock-down gave efficient (and probably more reproducible) results according to Figure 1B-C, why not using the same approach for analyzing loss-of-function phenotypes?(4) Similarly, this new paragraph regarding the knock-out strategy by Crispr is problematic: "However, in the Cad96ca mutant, 86% of the larvae (an editing efficiency of 67% by TA clone analysis) had a shortened feeding stage in the sixth instar and entered the metamorphic molting stage earlier, showing early pupation, with the pupation time being 24 h earlier. In the Fgfr1 mutant, 91% of the larvae (an editing efficiency of 61%) had a shortened feeding stage in the sixth instar and entered the metamorphic molting stage earlier, showing early pupation, with the pupation time being 23 h earlier" (lines 225-230).- How does the editing efficiency relate to the mutation efficiency few lines earlier (not clearly explained in the methods)? Were the animals homozygous or heterozygous for the mutations? - A shortened feeding stage can only be invoked if previous developmental transitions are unaffected. Such statement should be supported by a better description of the developmental timing phenotype (as suggested already by reviewer 2).

Thank you for your questions in (3) and (4). The editing rates of 67% and 61% for Cad96ca and Fgfr1 in individuals were calculated from the PCR products, indicating that the cells were mosaics by CRISPR/Cas9 editing. The mutants produced by CRISPR/Cas9 are mosaics. We removed the content to the methods section and increased the detail methods, Lines 705-717.

We increased discussion: "The phenotypes of gene mutation in *H. armigera* are somehow different from those obtained by homozygous mutation in other animals, due to the mosaic mutation by CRISPR/Cas9. In addition, RNAi of *Cad96ca* and *Fgfr1* was observed precocious pupation as was the case in CRISPR/Cas9, suggesting the RNAi can be used for the study of gene function in insect, especially when the gene editing is embryonic lethal". Lines 367-380.

We removed the improper description of the phenotypes in the results, such as that of the feeding stage. By increasing experiments of editing *Met1* individually (single knockout), *Cad96ca* and *Fgfr1* together (double knockout), and *Met1*, *Cad96ca* and *Fgfr1* together (triple knockout) to define the physiological roles of these receptors in JH signaling in vivo.

(5) Importantly, I don't understand where the new version of the figure 4E stems from. The « pupation on time » (blue) category present in the first version of the figure has now disappeared for mutant animals. Why? In the first, my understanding was that, among the mutant animals, around 50% had precocious pupation. In the new version of the figure 4E, the "pupation on time" category is missing, and the percentages of early pupation are therefore strongly increased... The explanations provided in the text are not clear regarding the reanalysis of the mutant phenotypes. In the first version of the manuscript, the following explanation was given: "In 61 survivors of Cas9 protein and Cad96ca-gRNA injection, 30 mutants were identified by the earlier pupation and sequencing (an editing efficiency of 49.2%)". Were all animals sequenced, or only the 30 displaying earlier pupation? Were the 31 others not sequenced or did they have no mutation? Could it be, as suggested by the first version of the figure, that some mutant animals did not display early pupation? It was indeed stated in the text that: "CRISPR/Cas9 editing by Cad96ca-gRNA or Fgfr1-gRNA injection resulted in earlier pupation (Figure 4D) for about 23-24 h by comparison with normal pupation in 46% and 54% of larvae, respectively". This new version of the figure should be explained.

Thank you for your reminder. The phenotype of pupation on time appeared in the first version, because we counted the phenotypes of all the surviving individuals injected with gRNA, that is, the survivors in Figure 4C, which including mutated and non-mutated individuals. According to the comments from first round of reviewers, we realized that it was inappropriate to count all the surviving individuals injected with gRNA, since there is no phenotype of pupation on time in the mutants. Therefore, in the second version, we replaced the picture by counting the phenotypes of all successfully mutated individuals, namely the mutants in Figure 4C.